# Müller cell glutamine metabolism links photoreceptor and endothelial injury in diabetic retinopathy

Katia Corano Scheri[1], Yi-Wen Hsieh[1], Thomas Tedeschi[1], James B Hurley[2], Amani A Fawzi[1]

We characterized the timeline of molecular dysfunction in diabetic retinopathy (DR) and diabetic retinal disease (DRD) by studying the streptozotocin (STZ)-induced mouse retina over the course of 6 mo of diabetes. We performed bulk RNA-Seq on endothelial and retinal cells, separately, at 1, 3, and 6 mo of diabetes and single-cell RNA-Seq (scRNA-Seq) at 3 months. Transcriptomics changes were validated by in vitro and ex vivo assays and immunohistochemistry of mouse and human tissue. Bulk RNA-Seq revealed inflammation in endothelial cells at 1 mo. At 3 mo, scRNA-Seq identified glutamine-driven anaplerotic dysfunction in Müller cells, confirmed by retinal culture. We posited this glutamine deficiency would impact the photoreceptors and endothelial cells. We validated this hypothesis using endothelial cells in vitro, and immunohistochemistry of disrupted photoreceptor ribbon synapses in mouse and human diabetic retinas. In addition, glutamine deprivation increased the expression of apoptotic genes in endothelial cells. At 6 mo, we observed significant down-regulation of angiogenic pathways and elevated profibrotic markers. Our results suggest that dysfunction of the metabolic ecosystem linking the Müller–photoreceptor–endothelial cells is central to the early stages of DRD pathogenesis, impacting photoreceptor synapses and endothelial cells, before the appearance of the classic microvascular features of DR.

## Introduction

Diabetic retinopathy (DR), the leading cause of blindness in American adults, is a microvascular complication of diabetes mellitus that leads to vision-threatening retinal complications (1, 2, 3, 4).

Although DR is clinically classified based on the vascular lesions (5, 6, 7), mounting evidence points to retinal neuronal dysfunction in the diabetic microenvironment that can precede the microvascular manifestation of DR (8). Together, DR and diabetic retinal neurodegeneration (DRN) constitute diabetic retinal disease (DRD), with a surging interest in unraveling the mechanisms that drive DRN (9, 10, 11, 12, 13, 14). The relative lack of emphasis on DRN thus far is puzzling, considering the retina is in fact an intricately organized neural tissue, with neurons interconnected by synapses and interneurons, supported by macroglia and the dual blood supply from the retinal and choroidal vasculature (15, 16). The retina is one of the most energy-consuming tissues, with the photoreceptors having the highest oxygen demand in the human body (17, 18). The retina preferentially uses glucose as its main energy substrate, with the photoreceptors consuming the majority of it (19, 20, 21, 22). Glucose is delivered from the choriocapillaris through the RPE to photoreceptors, and via retinal vessels to inner neurons (23) with a very complex, finely regulated crosstalk and metabolic ecosystem to exchange metabolites between the retinal cells (24). We reasoned disruption of this ecosystem by diabetes may underlie both neurodegenerative and microvascular changes in DRD.

To recapitulate DRD, we used the streptozotocin (STZ)-induced diabetes model (25). Early DRD features include gliosis (4–5 wk), ganglion cell loss (6 wk), nuclear layer thinning (10 wk), and vascular degeneration after 6 mo (25, 26, 27). To study the consequences of diabetes on the entire retina, we performed transcriptomics analyses during the first 6 mo of diabetes in the STZ mouse model to understand the pathways that modulate the vascular and neural degeneration in diabetes. We identified metabolic dysfunction of the glutamine–glutamate pathway as linking the Müller glia, photoreceptors, and vascular endothelial cells, providing important insights into neurodegeneration and vascular degeneration in DRD.

## Results

### Study design, overview, and differential gene expression analysis

To understand the molecular mechanisms of DRD, we used the STZ mouse model and assessed bulk transcriptomics at 1, 3, and 6 mo of diabetes (Fig 1A). Triplicates were included, excluding three degraded endothelial samples. Endothelial cells were isolated

[1]Department of Ophthalmology, Feinberg School of Medicine, Northwestern University, Chicago, IL, USA    [2]Department of Biochemistry, University of Washington, Seattle, WA, USA

Correspondence: afawzimd@gmail.com

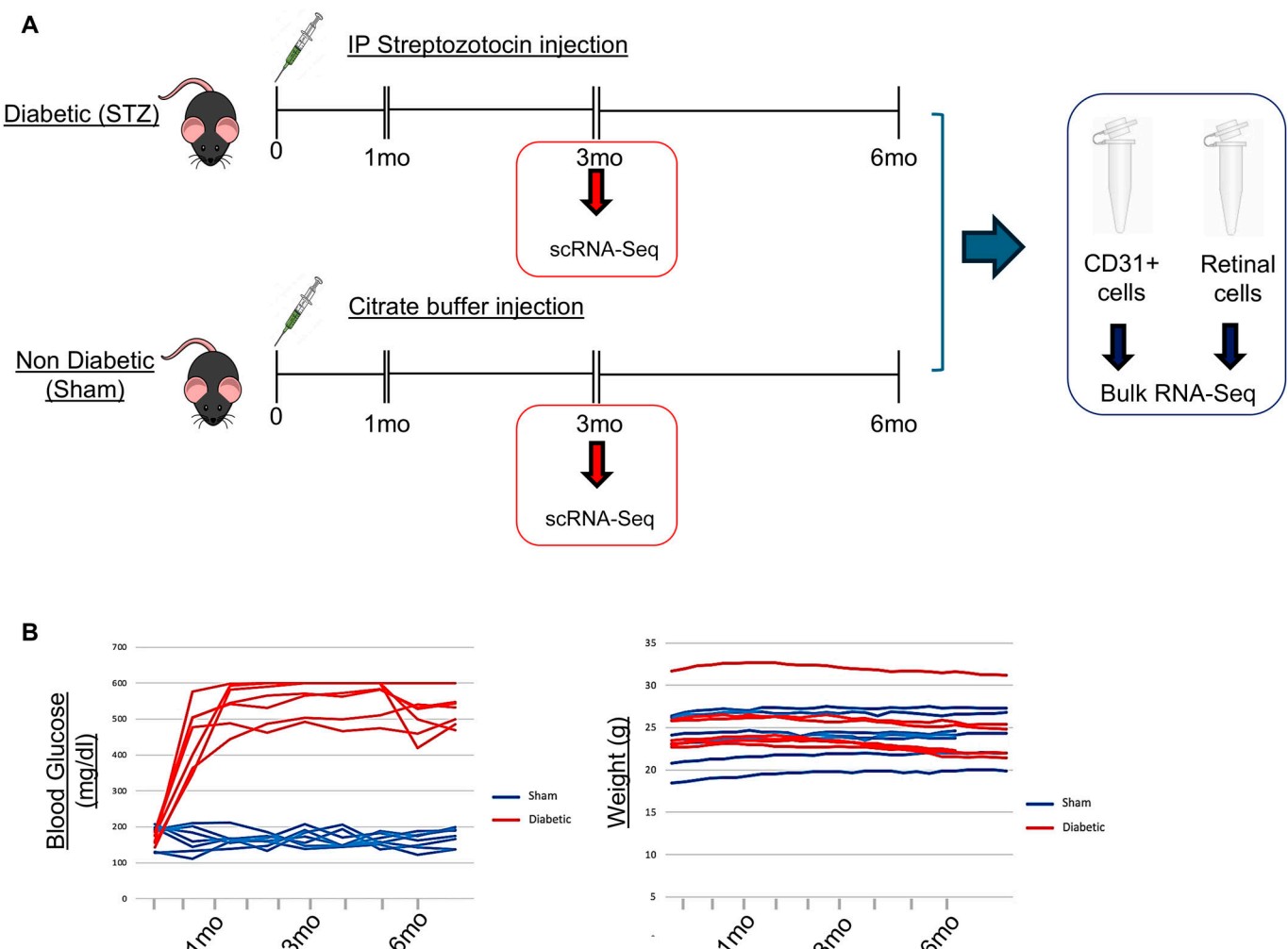

**Figure 1. Experimental plan and diabetes readout.**
**(A)** Schematic representation of the experimental design, time points, and transcriptomics analyses (bulk RNA-Seq and scRNA-Seq). **(B)** Blood glucose quantification (left) and body weight (right) over time in sham (blue line) and STZ (red line) mice are shown in the graphs.

using CD31-conjugated magnetic beads for bulk RNA-Seq of the endothelial and retinal compartments. In addition, we performed single-cell RNA-Seq (scRNA-Seq) at 3 mo using pooled retinas from six sham and six STZ mice (Fig 1A). Diabetes induction was confirmed by blood glucose and body weight monitoring (Fig 1B), and DR hallmarks were validated, including leukostasis, vascular leakage, pericyte dropout, and vascular compromise at respective time points (Fig S1A–C).

Bulk RNA-Seq analysis showed distinct gene expression profiles in diabetic versus control samples. At $|\log_2FC| > 1$ or $< 1$ and adjusted $P$-value $< 0.05$, we identified differentially expressed genes in the endothelial compartment: 160 at 1 mo, 60 at 3 mo, and 106 at 6 mo. In the retinal compartment, 155 genes were differentially expressed at 1 mo, 26 at 3 mo, and 254 at 6 mo (Table S1, Fig S2A). Time-course analysis identified consistently up-regulated ($|\log_2FC| > 0.5$) or down-regulated ($|\log_2FC| < -0.5$) genes. Pathway analysis conducted on these genes showed endothelial down-regulation of adherens junction genes and up-regulation of ion/

metal transport pathways, whereas in the retina, up-regulated genes were linked to neurotransmitter/glutamate receptor activity, and down-regulated genes to integrin-beta signaling (Table S2, Fig S2B).

## Endothelial cell inflammation and early impairment of retinal cells at 1 mo

We used differential gene expression analysis of bulk RNA-Seq to examine molecular changes over time. At 1 mo (STZ versus sham), the top 50 up- and down-regulated genes in endothelial cells are shown in the heatmap (Fig 2A). Gene Ontology (GO) analysis revealed up-regulation of inflammatory pathways, particularly IFN-γ/β responses and MHC class II processes (Fig 2B). MHC class II molecules, such as H2A, are heterodimer molecules usually expressed on the surface of professional antigen-presenting cells of the immune system, and up-regulated by inflammatory cells during an infection or inflammatory state (28). MHC molecules on

## Endothelial cells

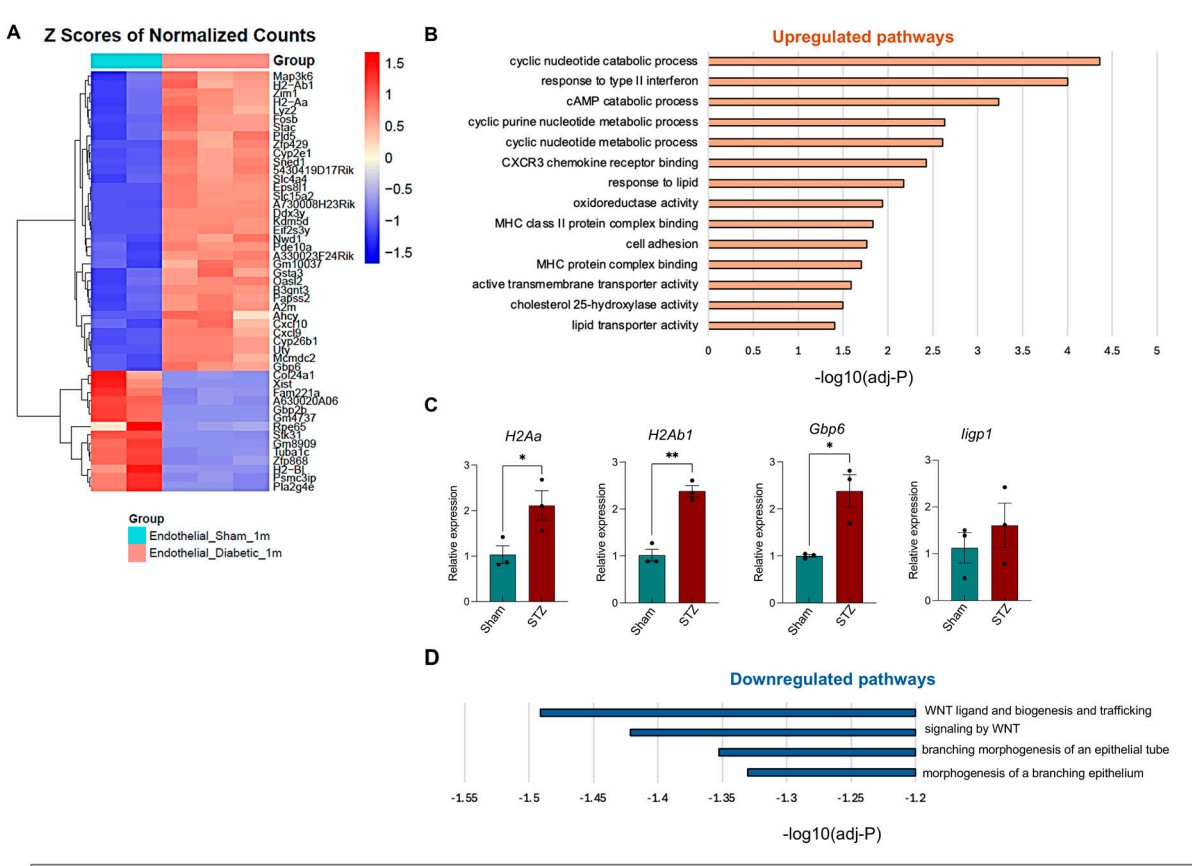

## Retinal cells

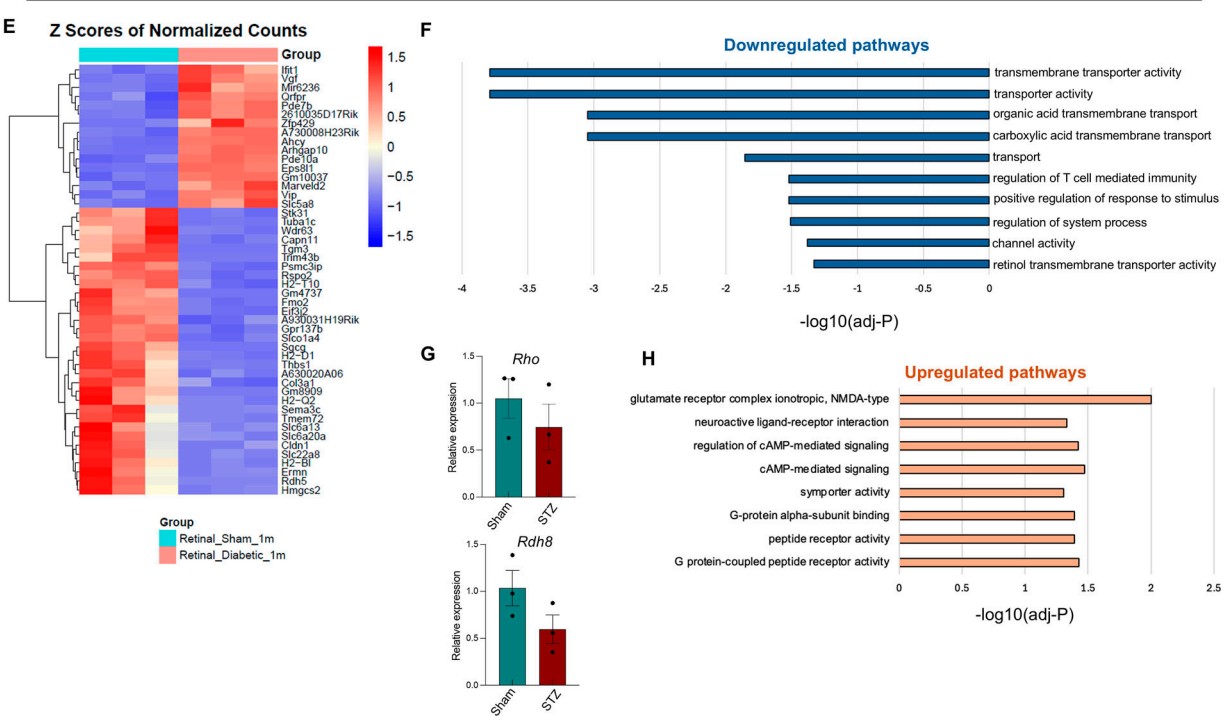

the surface of endothelial cells can be recognized by immune cells, triggering trans-endothelial migration and activation (29). Interferon-inducible genes such as *Gbp6* and *Ilgp1* are associated with immune signaling and inflammation (30). Using qRT-PCR on endothelial cell-enriched lysates, we confirmed a significant increased expression of these pathway-related genes, such as *H2Aa*, *H2Ab1*, *Gbp6*, and a trend for *Ilgp1* (Fig 2C). Down-regulated pathways included epithelial morphogenesis, differentiation, and WNT signaling (Fig 2D), indicating early endothelial dysfunction at this early stage of DR. The WNT pathway is known to be involved in cellular proliferation, survival, differentiation, migration, and apoptosis and plays an important role in angiogenesis and vessel remodeling (31, 32, 33, 34). The role of the WNT pathway in development and differentiation of the retinal microvasculature and blood–retinal barrier has been described previously (34, 35, 36, 37, 38, 39), and WNT signaling is activated in the early stages of retinal vascular development (34, 39, 40, 41). Abrogation of retinal WNT signaling in mouse models induced maldevelopment of intraretinal capillaries, defects in vascular patterning and branching, and intraocular hemorrhages (42, 43). Literature data show also that a missense mutation in LRP6, which encodes a coreceptor in the WNT signaling pathway, impairs WNT signaling resulting in coronary artery diseases and multiple cardiovascular risk factors (44). Moreover, activation of WNT signaling normalizes retinal vasculature in a mouse model of familial exudative vitreoretinopathy (45). In addition, alteration of WNT signaling molecules might be associated with diabetes onset (46, 47).

In the retina, differentially expressed genes showed down-regulation of transmembrane transporter activity, channel activity, and retinol transmembrane transporter activity (Fig 2E and F), suggesting disrupted homeostasis. To investigate the retinol pathways, crucial for photoreceptor function, we analyzed the expression of rod-specific retinoid cycle genes *Rho* and *Rdh8* by qRT-PCR and found they were slightly reduced in STZ retinas (Fig 2G), indicating the onset of impaired scotopic vision. Rhodopsin, encoded by the *Rho* gene, is a G protein–coupled receptor and the most abundant protein in rod photoreceptors. It functions as the primary molecule of scotopic vision, and its activation by light photons is followed by a small, graded hyperpolarization in membrane potential, thus initiating the phototransduction process (48). A reduction of rhodopsin content has been reported in the rat model of DR at 4 mo after diabetes induction, along with a significant reduction in both scotopic and photopic amplitudes of the a- and b-waves of the electroretinogram (ERG) responses (49). Decreased rhodopsin and 11-cis-retinal levels have also been described in diabetic

mice after 4 mo of hyperglycemia (50). *Rdh8* encodes one of the retinol dehydrogenases (RDH) involved in the visual cycle, localized in the photoreceptor outer segments where it reduces all-trans-retinal to all-trans-retinol (51).

Up-regulated pathways included glutamate receptor complex NMDA type, symporter activity, cAMP signaling and G protein–coupled receptor activity (Fig 2H). Glutamate receptors, NMDA type, are known to play a crucial role in processes such as visual signal transmission and the modulation of inhibitory feedback (52). Their up-regulation at this stage might be a response to the altered homeostasis in the retina. G protein–coupled receptors (GPCRs) are integral membrane proteins that associate with heterotrimeric G proteins and induce signaling pathways when activated by a variety of ligands such as ions, proteins, neurotransmitters, and hormones (53). G protein–coupled receptors may be up-regulated in the diabetic retina as part of the retina's response to the metabolic, inflammatory, and oxidative stress associated with diabetes. Previous studies have shown activation of GPCR91 in DR and hypoxic retinal diseases, with the consequent activation of VEGF pathways and breakdown of the blood–retinal barrier (54, 55), which is one of the molecular features of DR.

### Classification of cell clusters from single-cell RNA-Seq at 3 mo in diabetic and control retinas

We were puzzled by the relatively low number of dysregulated genes in bulk RNA-Seq at 3 mo of diabetes; therefore, we performed scRNA-Seq analysis at this time point. Using canonical markers, we identified and labeled major retinal cell types (UMAP, Fig S3A; dotplot, Fig S3B). Analysis of cell-type proportions using hypergeometric distribution revealed uneven representation between sham and STZ retinas (Fig S3C). This may reflect biological differences because of diabetes or differential sensitivity to tissue digestion. To assess gene expression changes, we reclustered each cell type individually.

### Alterations of ribbon synapse gene expression and morphology in diabetic retinas at 3 mo

When we reclustered cones and rods separately, we identified 5 clusters for each cell type (Fig 3A and C). Gene expression analysis revealed significant down-regulation of *Cacnb2*, in both cones and rods (Fig 3B and D). *Cacnb2* encodes the β2 subunit of the voltage-gated $Ca^{2+}$ channel (Cavβ2) in photoreceptor ribbon synapses. Ribbon synapses are specialized synapses of the vertebrate retinal photoreceptors, critical for transmitting the

---

**Figure 2. Endothelial and retinal cell gene expression changes at 1 mo in bulk RNA-Seq.**
**(A)** Heatmap of the top 50 up-regulated (red) and down-regulated (blue) genes in endothelial cells from sham (cadet blue) and diabetic (STZ; red) retinas. **(B)** Pathway enrichment analysis using Metascape, comparing the up-regulated genes in STZ endothelial cells with sham endothelial cells. **(C)** Validation by qRT-PCR of *H2a2, H2ab1, Gbp6,* and *Iigp1* gene expression in enriched endothelial cells from control and diabetic retinas. *Acta2* was used as a housekeeping gene. The Mann–Whitney *U* test was used for statistical analysis. *$P < 0.05$, **$P < 0.01$. sham (n = 3); STZ (n = 3). **(D)** Pathway enrichment analysis of the down-regulated genes in STZ compared with sham endothelial cells using Metascape. **(E)** Heatmap of the top 50 up-regulated (red) and down-regulated (blue) genes comparing cells isolated from sham (cadet blue) and STZ (red) retinas. **(F)** Pathway enrichment analysis of the down-regulated genes in STZ compared with sham retinas using Metascape. **(G)** Validation by qRT-PCR of *Rho* and *Rdh8* gene expression in healthy and diabetic retinas. *Acta2* was used as a housekeeping gene. The Mann–Whitney *U* test was used for statistical analysis. sham (n = 3); STZ (n = 3). **(H)** Pathway enrichment analysis of the up-regulated genes in STZ retinas compared with sham retinas using Metascape is shown. Source data are available for this figure.

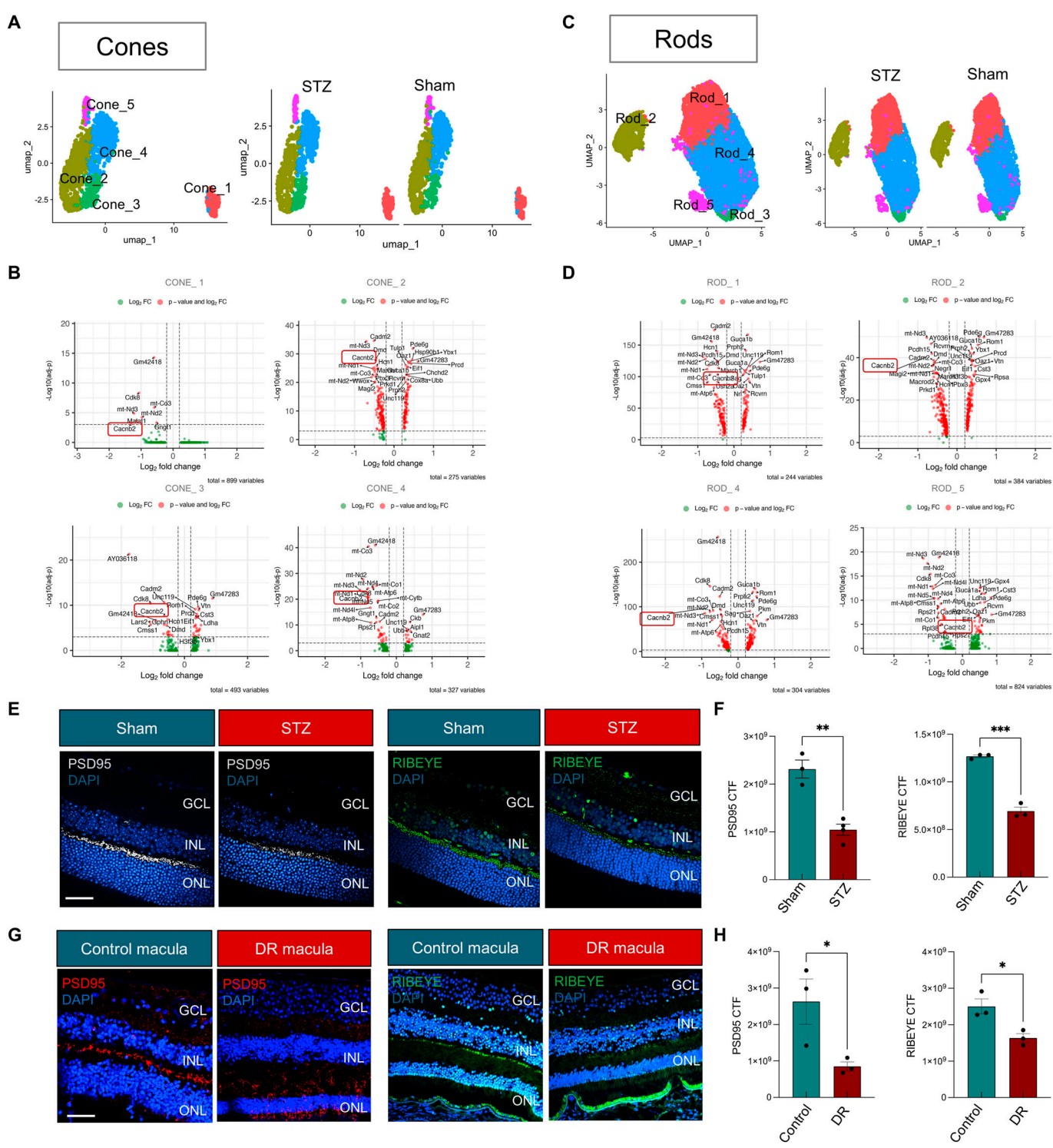

**Figure 3. Photoreceptor clustering and gene expression changes at 3 mo from scRNA-Seq.**
(A) Representative UMAP plot of the different cone clusters (combined samples versus separate sham and STZ) is shown. (B) Representative volcano plot of the differentially expressed genes in STZ compared with sham cones, showing down-regulation of *Cacnb2*. Dashed lines indicate thresholds for significance: |log$_2$FC| > 0.25 or <−0.25 and −log$_{10}$ (*P*-value) ≥ 3. (C) Representative UMAP plot of the different cluster of rods (all samples together and split in sham and STZ) is shown. (D) Representative volcano plot of the most differentially expressed genes in STZ compared with sham rods, showing down-regulation of *Cacnb2*. Dashed lines indicate thresholds for significance: |log$_2$FC| > 0.25 or <−0.25 and −log$_{10}$ (adj *P*-value) ≥ 3. (E) Immunofluorescence on 3-mo retinal cross-sections for PSD95 (left) and RIBEYE (right). DAPI was used to counterstain the nuclei. Retinal layers are labeled. Scale bars: 50 $\mu$m. (F) Quantification of corrected total fluorescence (CTF) for both proteins in mouse samples is shown in the graphs. (G) Representative images of immunofluorescence for PSD95 and RIBEYE on paraffin-embedded histological cross-sections of

graded light responses to the second-order bipolar and horizontal cells (56). Unlike nonribbon synapses, voltage-gated L-type Ca2+ channel opening at ribbon synapses triggers a multiquantal release that can be highly synchronous; therefore, Ca2+ influx through voltage-gated Ca2+ channels triggers the release of neurotransmitters at the presynaptic terminal (57). To evaluate whether *Cacnb2* down-regulation affected the ribbon synapse morphology in the diabetic mouse, retinal cross-sections were stained for PSD95 (a postsynaptic scaffolding protein) and RIBEYE (the main structural component of the presynaptic ribbon), and we detected a significant reduction in both protein intensity of fluorescence and area in STZ mice compared with controls (Figs 3E and F and S4A). This decline was not observed at 1 mo of diabetes, but progressed from 3 to 6 mo of diabetes (Fig S4B and C). Human donor eyes with diabetic retinopathy (DR) exhibited similar disorganization and reduction in PSD95 and RIBEYE in the macular outer plexiform layer (OPL) when compared to healthy controls (Fig 3G and H), and no substantial changes were observed in other synapse proteins, such as EAAT5 (Fig S4D).

Our scRNA-Seq data revealed in both cones and rods a significant down-regulation of *Slc38a3*, encoding a transporter for glutamine, critical for photoreceptor metabolism and function (Fig 4A). Glutamine is physiologically produced by glial cells, especially Müller cells, and shuttled into the photoreceptors, where it is converted into glutamate, the main excitatory neurotransmitter in the photoreceptor ribbon synapses, and an important molecule for photoreceptor metabolism and survival (58). Consistent with the transcriptomics data, Western blot analysis confirmed a significant reduction of the SLC38A3 protein in STZ retinas (Fig 4B). In addition, we performed immunofluorescence for SLC38A3 and we found a significant reduction in the inner segment (IS) of photoreceptors in the STZ retinas when compared to sham (Fig 4C).

To further characterize the molecular changes at this time point, we analyzed bulk RNA-Seq data at 3 mo. Pathway enrichment analysis revealed up-regulation of pathways associated with the synaptic membrane, presynaptic function, and apoptosis, suggesting ongoing retinal dysfunction. Notably, glycinergic synaptic transmission was significantly up-regulated, indicating a possible compensatory or maladaptive shift in the excitatory/inhibitory balance in response to early neuronal stress in diabetes (Fig 4D, top). In contrast, pathways related to glutamate receptor binding were down-regulated, supporting our scRNA-Seq and immunofluorescence findings and pointing to progressive photoreceptor decline. We also observed down-regulation of pyruvate and lactate transport pathways, consistent with metabolic impairment in the diabetic retina (Fig 4D, bottom). Our data suggest that photoreceptors in diabetes suffer from impaired glutamine uptake, affecting glutamate production and metabolism, and, together with ribbon synapse morphology alteration, suggest that ribbon synapse signal transmission might be impaired.

## Anaplerotic switch in Müller cells at 3 mo

Müller cells are unique retinal macroglia, oriented vertically across the entire retina with extensive processes that actively interact with, and support all retinal cells and the vasculature. We reclustered Müller cells and found four clusters (Fig 5A and B), and detected significant down-regulation of electron transport chain (ETC) genes (*mt-Nd1*, *mt-Nd2*, *mt-Nd3*) in cluster MC_3 suggesting metabolic alterations, and an up-regulation of vimentin (*Vim*) in this cluster, supporting the hypothesis of the initial stage of cell phenotype change. In addition, two other genes, *Glul* and *Slc1a3*, were significantly down-regulated in the diabetic Müller cells (MC_4) (Fig 5C). The *Glul* gene codifies glutamine synthase (GS), the enzyme responsible for glutamine production from glutamate (59). *Slc1a3* encodes GLAST (or EAAT1), the Müller cell glutamate transporter that uptakes excess extracellular glutamate, excitotoxic to neurons, thereby detoxifying the retinal microenvironment (60, 61, 62). Glutamate is mostly recycled at the synaptic level, but GLAST is responsible for preventing the lateral spread of the neurotransmitter by uptaking excess glutamate that diffuses out of the synaptic clefts. Müller cells then convert this "toxic" glutamate (via GS) into "nontoxic" glutamine, which is then transported extracellularly to support other retinal cells, including photoreceptors and endothelial cells (63, 64). Western blot of whole retinal protein lysates showed a trend in down-regulation of GS in the STZ samples, and no significant change in GLAST expression (Fig 5D). Although Müller cells are the main GS-expressing cells in the retina, GS and GLAST are expressed by other cells such as astrocytes (65). Therefore, we presume the protein quantification in whole retinal lysates may be confounded by contribution from other cells. We then performed immunofluorescence in retinal cross-sections and confirmed significant decrease in GS expression in the inner nuclear layer (INL) (at the Müller cell body level) in STZ when compared to sham retinas, as shown in Fig S5. We also stained the retinal cross-sections for CRALBP and GLAST; we observed a significant decrease in GLAST at the INL, but no changes in the expression and/or localization of CRALBP in diabetic retinas when compared to healthy ones, confirming selective decrease in GS and GLAST compared with other non–glutamine-related Müller cell markers (Fig S5).

To directly evaluate the effect of glucose on the glutamine–glutamate cycle in Müller cells, we cultured the spontaneously immortalized human Müller cell line, MIO-M1, in high glucose (HG) for 24 and 48 h, and compared that with cells cultured in normal glucose or mannitol (osmotic control). As shown in Fig 5E, we observed a significant reduction of glutamine in the cell media (released glutamine), as early as 24 h in high glucose, and this reduction was maintained at 48 h. We also quantified intracellular glutamine, which was significantly decreased after 48 h of high-glucose exposure. When we evaluated glutamate, we found the opposite effect, a significant increase as

the diabetic (DR) donor macula compared with control are shown. **(H)** Quantification of corrected total fluorescence (CTF) for both proteins in human samples is shown in the graphs. DAPI was used to counterstain the nuclei. Retinal layers are labeled. Scale bars: 50 $\mu$m.
Source data are available for this figure.

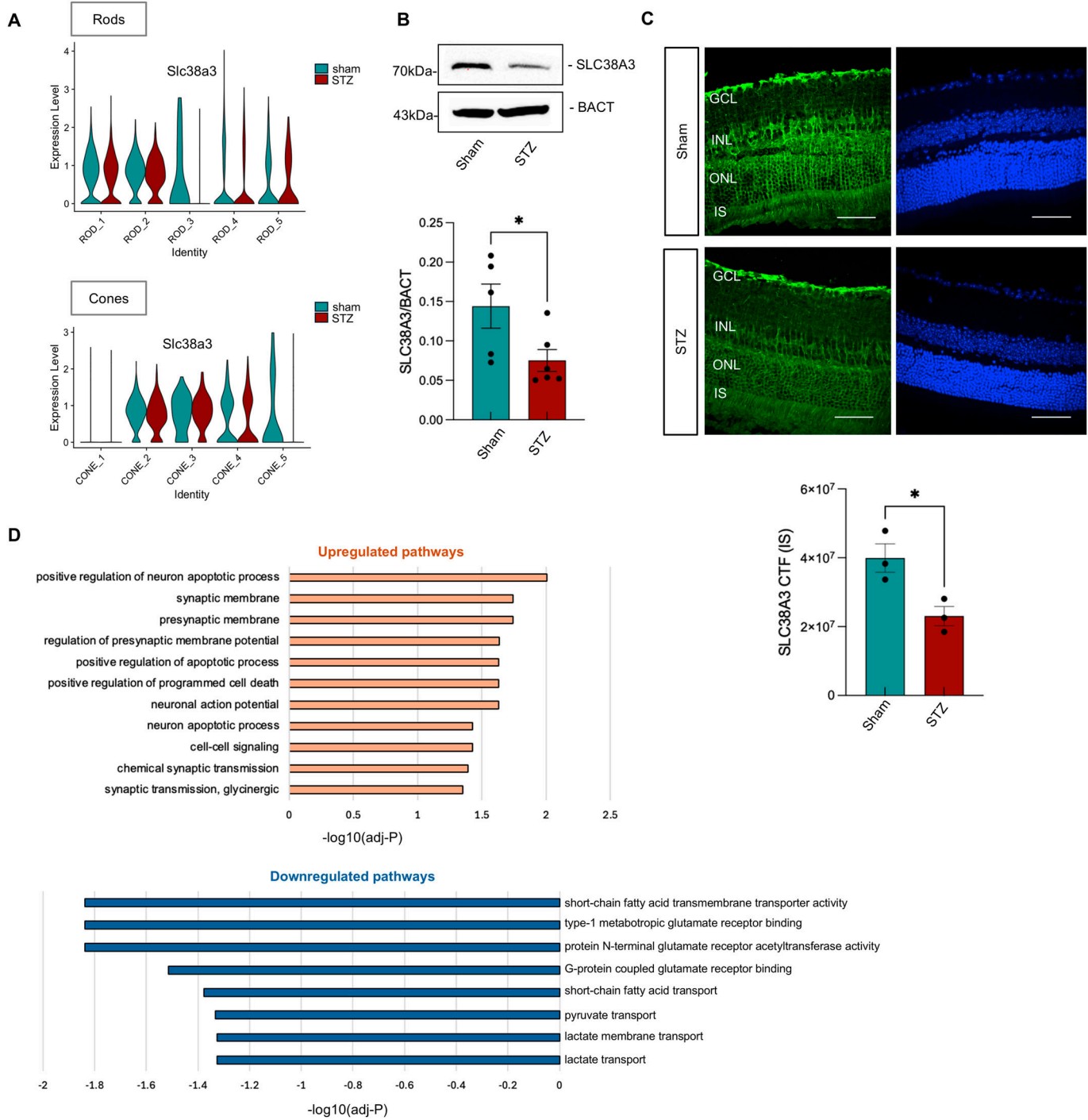

**Figure 4. Photoreceptor gene expression changes at 3 mo from scRNA-Seq and Bulk RNA-Seq.**
**(A)** Representative violin plots showing the expression of *Slc38a3* in the different rod (top) and cone (bottom) clusters analyzed by scRNA-Seq. **(B)** Western blot analysis for SLC38A3 on whole retinal lysates in 3-mo sham and STZ retinas. Representative bands are shown. Band densitometric analysis results are shown in the graphs. The Mann–Whitney *U* test was used for statistical analysis. *$P < 0.05$. sham (n = 5); STZ (n = 6). β-Actin (BACT) was used as a loading control. **(C)** Representative images of SLC38A3 staining on cross-sections of 3-mo STZ murine eyes compared with sham eyes. DAPI was used to counterstain the nuclei. Retinal layers are labeled. Scale bar: 50 μm. The quantification of corrected total fluorescence (CTF) is shown for the IS (inner segment) of photoreceptors. **(D)** Pathway enrichment analysis of the up-regulated (top) and down-regulated (bottom) genes from 3-mo bulk RNA-Seq data in STZ retinal cell compartment compared with sham is shown. Source data are available for this figure.

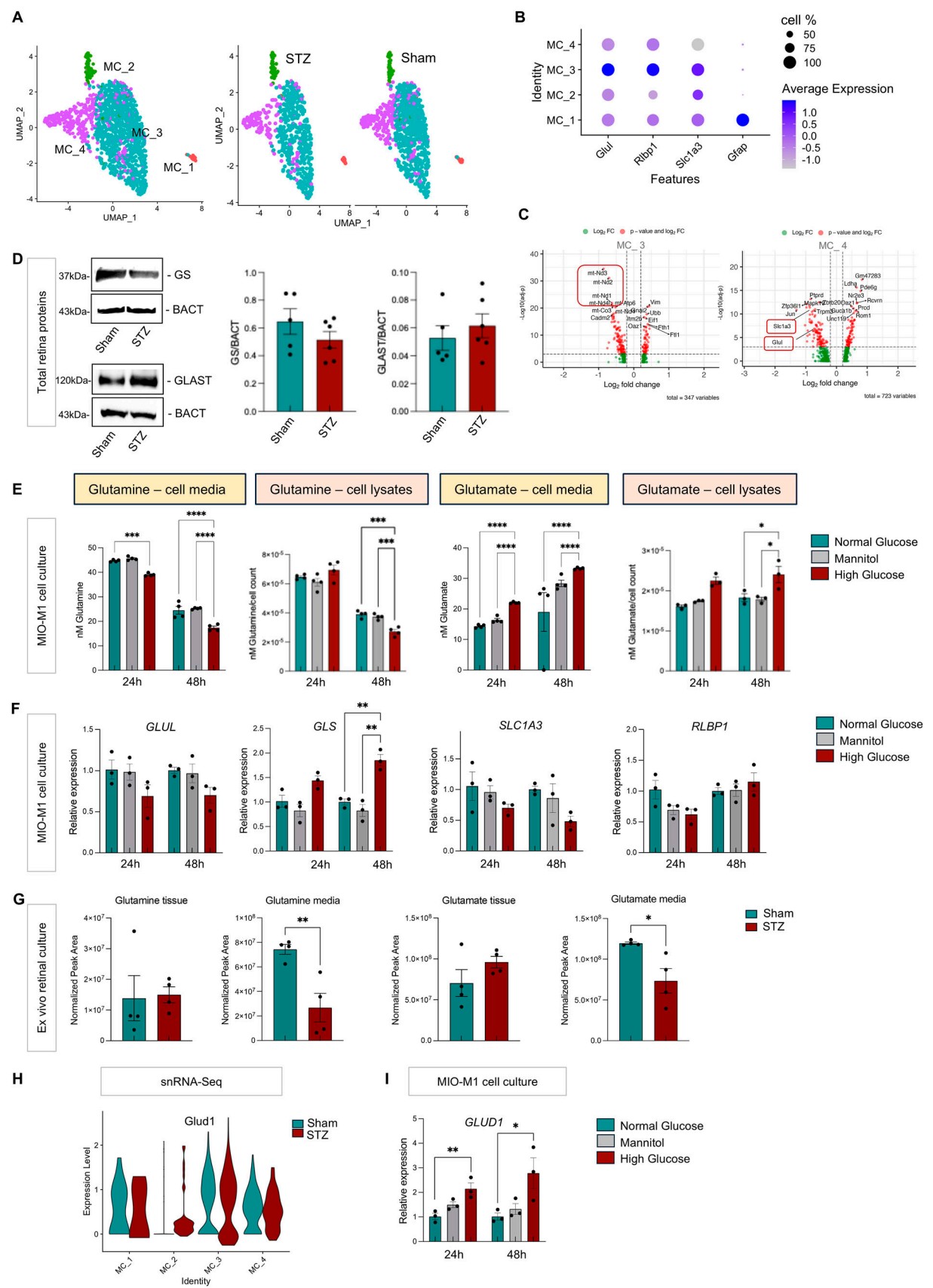

early as 24 h in the media, and intracellularly after 48 h, indicating Müller cells are not uptaking the glutamate in the extracellular environment. The increased production of glutamate intracellularly is likely from intracellularly retained (unreleased) glutamine.

We next analyzed the expression of the enzymes that regulate the glutamine–glutamate cycle by qRT-PCR, and observed a trend of down-regulation of *GLUL*, responsible for glutamine production, at both time points, and a significant up-regulation of *GLS*, which transforms glutamine into glutamate (Fig 5F) after 48 h. Moreover, we observed a trend for down-regulation of *SLC1A3*, responsible for uptake of excess glutamate from the extracellular environment. To explore whether glucose disrupts other metabolic support functions of Müller cells, we analyzed the expression of *RLBP1* (that codifies for CRALBP), the enzyme providing the 11-cis-retinal photosensitive chromophore to cones, and found no significant change in response to glucose (Fig 5F). Based on these data, we hypothesize that in the setting of high glucose, Müller cells switch to anaplerosis, using intracellular glutamine for their own energetic needs, and instead of removing the glutamate from the environment and releasing it as glutamine to support retinal endothelial and neuronal energy production and neurotransmission respectively, they allow glutamate to accumulate in the extracellular environment with potential detrimental effects on the retinal neurons. We are aware that these immortalized cultured M0-M1 cells are likely to behave differently compared with Müller cells in their in situ retinal environment; therefore, we sought to further validate these results. We performed ex vivo whole retinal culture of 3-mo diabetic retinas and their controls (58, 66), cultured with aspartate and lactate, the substrates for glutamine synthesis by Müller cells in vivo. We quantified the glutamine–glutamate metabolites by mass spectrometry in the retinal tissues and culture media. As shown in Fig 5G, we observed a significant reduction of glutamine in the culture media of STZ retinas, supporting the hypothesis that diabetic Müller cells make less glutamine available to the other retinal cells, thereby depriving them of this amino acid. Our data also indicated a trend of increased glutamate levels in the retinal tissue supporting the accumulation of potentially "neurotoxic" glutamate in the retina. We observed a significant reduction of glutamate in the culture media of STZ-cultured retinas, likely indicating that other retinal cells are taking up the excess glutamate from the environment,

especially under conditions where GLAST expression is reduced in Müller cells. As mentioned above, under normal conditions Müller cells normally uptake the excess extracellular glutamate (released by inner retinal neurons) using GLAST transporter and convert it into glutamine, which is then shuttled back into neurons to maintain homeostasis. Under the diabetic condition, Müller cell dysfunction and reduction of GLAST expression might impair this uptake.

## Metabolic pathway alterations in Müller cells and photoreceptors at 3 mo

We investigated how changes in the glutamine–glutamate cycle in Müller cells affect their energy pathways and neighboring retinal cells. We observed increased *Glud1* expression in one Müller cell cluster in diabetes (Fig 5H). GLUD1 converts glutamate into alpha-ketoglutarate ($\alpha$KG), a key intermediate in the TCA cycle. Normally, Müller cells convert lactate from photoreceptors and endothelial cells to pyruvate for the TCA cycle, also regulating retinal pH (58). qRT-PCR analysis on MIO-M1 exposed to high glucose also confirmed the up-regulation of *GLUD1* after 24 h of glucose exposure, further evidence of the anaplerotic switch within Müller cells (Fig 5I). Considering the scenario where Müller cells rely on their own metabolites for their metabolic needs, we analyzed the expression of glycolytic enzymes such as *Pkm* (pyruvate kinase), which is normally expressed at low levels in Müller cells (58), *Ldha* (lactate dehydrogenase A), *Pfkm* (phosphofructokinase), and *Hk1* (hexokinase 1). We found a significant increased expression of *Pkm* and *Ldha* (Fig S6A) in one cluster of Müller cells. These results suggest that some Müller cells attempt to up-regulate glycolysis to support their own glucose metabolism. We observed also a significant increase of *Pkm* and *Ldha* in two clusters of cones and almost all the rods (Fig S6B and C). Our data show that photoreceptors, particularly rods, appear to increase glucose use in response to glutamine deprivation under hyperglycemia.

Overall, cell culture, ex vivo whole retinal experiments, and scRNA-Seq data suggest that Müller cells undergo an anaplerotic switch in high glucose, with the unintended outcome of starving other cells from glutamine, disrupting the critical metabolic ecosystem that supports retinal neurons and endothelial cells.

**Figure 5. Müller cell clustering and gene expression changes at 3 mo from scRNA-Seq.**
**(A)** Representative UMAP plot of the different cluster of Müller cells (combined samples versus separate sham and STZ) and **(B)** dotplot with the representative cell-type markers are shown. **(C)** Representative volcano plot with the top differentially expressed genes in STZ compared with sham Müller cells, showing down-regulation of mitochondrial genes and glutamine–glutamate cycle genes *Glul* and *Slc1a3*. Dashed lines indicate thresholds for significance: |log$_2$FC| > 0.25 or <−0.25 and −log$_{10}$ (adj *P*-value) ≥ 3. **(D)** Western blot analysis for glutamine synthase (GS) and GLAST (or SLC1A3, solute carrier family 1 member 3) on whole retinal lysates in 3-mo sham and STZ retinas. Representative bands are shown. Band densitometric analysis results are shown in the graphs. sham (n = 5); STZ (n = 6). $\beta$-Actin (BACT) was used as a loading control. **(E, F)** Intracellular and extracellular glutamine and glutamate quantification in MIO-M1 cells cultured in high glucose (30 mM) for 24 and 48 h, compared with normal glucose (5 mM) and mannitol, and (F) mRNA expression for glutamine–glutamate cycle genes upon 24 and 48 h in high-glucose (30 mM) culture is shown. *Acta2* was used as a housekeeping gene. A summary of three independent experiments is shown. One-way ANOVA followed by Tukey's multiple comparison test was used. *$P$ < 0.05, **$P$ < 0.01, ***$P$ < 0.001, ****$P$ < 0.0001. Normal glucose (n = 3); high glucose (n = 3); mannitol (n = 3). **(G)** Quantification of glutamine and glutamate by mass spectrometry in 3-mo sham (n = 4) and STZ (n = 4) retinas after culturing them in 5 mM aspartate and 5 mM lactate for 90′. The Mann–Whitney test was used for statistical analysis. **$P$ < 0.01. **(H)** Violin plots showing *Glud1* expression in each Müller cell cluster. **(I)** Validation by qRT-PCR of *GLUD1* expression in MIO-M1 cells cultured in high glucose for 24 and 48 h, compared with normal glucose and mannitol. *Acta2* was used as a housekeeping gene. A summary of three independent experiments is shown. One-way ANOVA followed by Tukey's multiple comparison test was used. *$P$ < 0.05, **$P$ < 0.01, ***$P$ < 0.001, ****$P$ < 0.0001. Normal glucose (n = 3); high glucose (n = 3); mannitol (n = 3).
Source data are available for this figure.

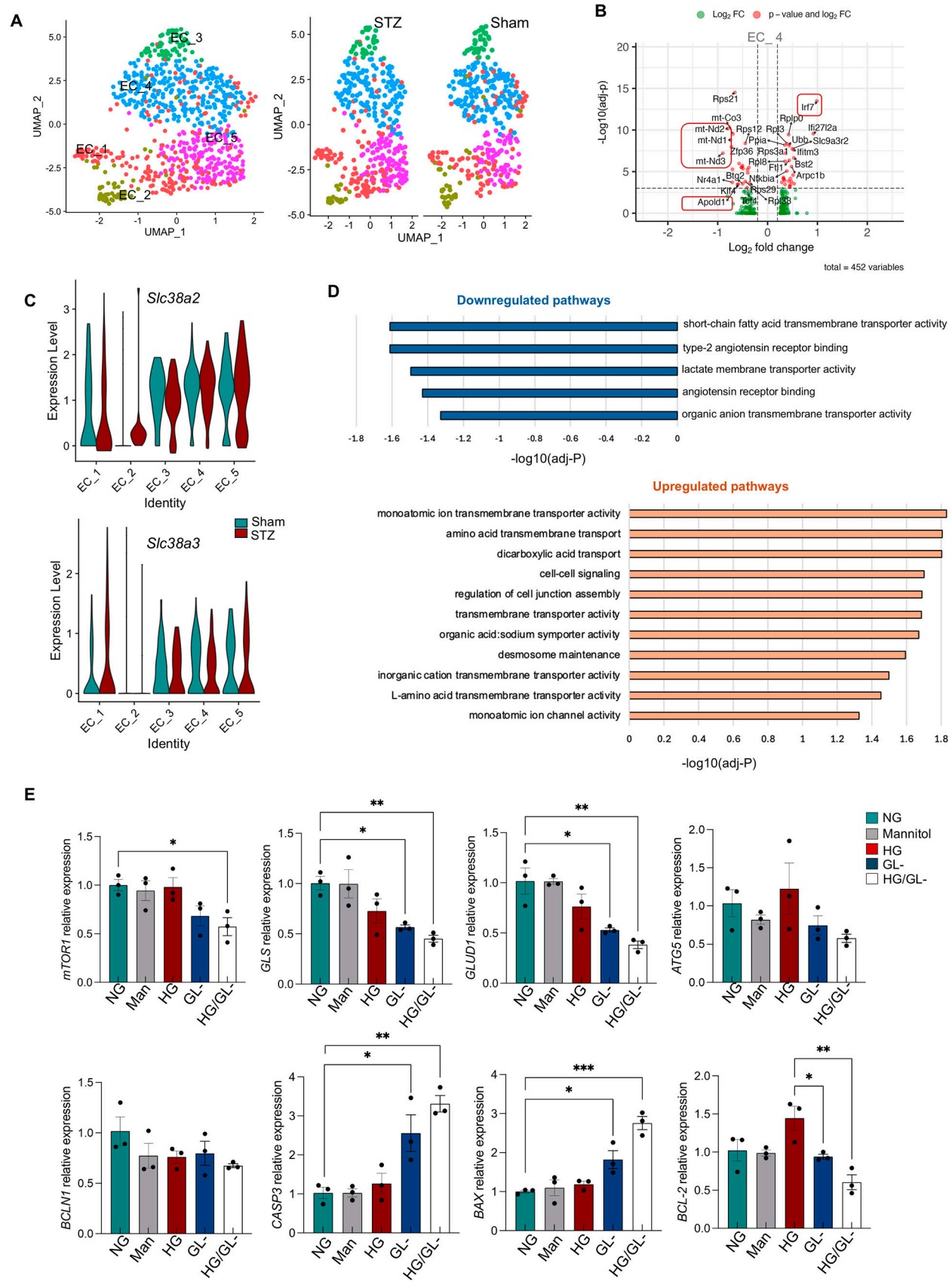

## Endothelial cell metabolic and inflammatory state at 3 mo

We next reclustered the endothelial cells (Fig 6A) and performed gene expression analysis, and observed down-regulation of mitochondrial genes, particularly electron transport chain (ETC) genes (*mt-Nd1*, *mt-Nd2*, *mt-Nd3*, *mt-Co3*) and *Apold1* (apolipoprotein L domain containing 1) (Fig 6B). This suggests altered endothelial cell metabolism, endothelial cell stability, and vessel homeostasis at this stage. Moreover, we found up-regulation of interferon pathway–related genes, such as *Irf7*, *Ifi27l2a*, *Ifitm3*, suggesting persistently activated inflammatory state of the endothelial cells and extending the observations in bulk RNA-Seq at 1 mo.

Because endothelial cells rely on glutamine, provided by Müller cells, for their metabolism (64), we wondered how they are impacted by the Müller cell anaplerotic switch. We did not observe any change in the glutamine uptake transporters, *Slc38a2*, *Slc38a3*, indicating surviving endothelial cells maintain expression of those genes and their ability to uptake the amino acid (Fig 6C). Interestingly, we observed no significant alterations in other metabolic pathways, including glycolysis, based on the expression levels of key glycolytic enzymes (Fig S6D). We wondered whether scRNA-Seq differentially under-sampled "diseased" endothelial cells, obscuring their gene expression changes. To address this, we performed pathway enrichment analysis on the endothelial-enriched bulk RNA-Seq data at 3 mo. To overcome the observed endothelial cell contamination with photoreceptor genes on bulk RNA-Seq, we focused our analysis on endothelial cell–specific gene signatures and excluded any photoreceptor-specific genes. Among the top down-regulated pathways, we identified fatty acid transporter activity, lactate transport, and type 2 angiotensin signaling, suggesting metabolic alterations and vascular signaling alteration (Fig 6D, top). In contrast, the top 10 up-regulated genes revealed changes in amino acid and ion transport, potentially reflecting a compensatory response to Müller cell glutamine deficiency. We also observed up-regulation of pathways involved in cell junction assembly and cell–cell signaling, likely indicating compensatory responses to the vessel leakage observed at this time point (Fig 6D, low). Considering that glutamine is critical for endothelial homeostasis and proliferation, as well their energy production and TCA cycle, we sought to explore how the Müller cell anaplerotic switch and subsequent endothelial glutamine deprivation affected the endothelial mTOR and other metabolic pathways (67). The literature suggests that mTOR, a central regulator of cell growth and metabolism, is a sensor of nutrients and amino acids (68), such as glutamine, and that in the absence of nutrients and amino

acids, mTOR is deregulated together with other metabolic pathways (69, 70, 71). We cultured human retinal microvascular endothelial cells (HRMEC) in high glucose (HG), with and without glutamine deprivation (GL-) for 48 h (Fig 6E). qRT-PCR analysis showed significant down-regulation of *mTOR1* in high glucose with glutamine deprivation (HG/GL-) and a trend of down-regulation in glutamine starvation alone (GL-). We also detected a significant reduction of *GLS* and *GLUD1* both in HG/GL- and in GL-, suggesting that glutamine deficiency interferes with the expression of these enzymes. GLS converts glutamine into glutamate, which is then converted into αKG by GLUD1 to fuel the TCA cycle. To better understand how glutamine deprivation affects endothelial cells, we next analyzed autophagic gene expression and found no significant alteration of *ATG5* and *BCLN1*. Instead, we observed significant up-regulation of pro-apoptotic genes, such as *BAX* and *CASP3*, and down-regulation of the anti-apoptotic gene *BCL-2* (Fig 6E) in HG/GL- and GL-. Our data suggest that glutamine deprivation in hyperglycemia down-regulates glutaminolysis gene expression in endothelial cells and that in response to these changes, the endothelial cells up-regulate apoptotic genes. The activation of apoptotic signaling likely contributes to endothelial cell loss and dysfunction. Notably, vascular integrity was impaired at this time point. As previously shown in Fig S1, the deep retinal layer exhibits a marked increase in vascular leakage, as demonstrated by the biocytin assay, indicating a substantial loss of vascular stability in diabetic retinas.

## Endothelial cell alterations and retinal fibrogenesis at 6 mo

Given the gene expression alterations in the endothelial cells at 3 mo and the leakage increase in the retinal deep layer, we were not surprised to find significant down-regulation of endothelial genes and pathways involved in blood vessel morphogenesis, vasculogenesis and angiogenesis, and cell adhesion and communication, all signs of endothelial dysfunction at 6 mo of diabetes, considered a late stage of DR in mouse models (Fig 7A and B). These data confirm that the alterations observed at earlier time points culminate in complete endothelial dysfunction by 6 mo. On the other hand, the retinal cell compartment showed significant up-regulation of pathways related to collagen biosynthesis and formation (Fig 7C and D, top). Among the down-regulated pathways, we found calcium ion binding, flippase activity, neurotransmitter activity, confirming a decrease in those important retinal functional pathways at this stage (Fig 7D, bottom).

The fibrogenic process was validated by qRT-PCR, showing significant up-regulation of *Col6a1* and *Ph42a* (Fig 7E). To further

**Figure 6. Endothelial cell clustering and gene expression changes at 3 mo from scRNA-Seq and Bulk RNA-Seq.**
**(A)** Representative UMAP plot of the different endothelial cell clusters (combined versus separate sham and STZ samples) is shown. **(B)** Representative volcano plot with the main differentially expressed genes in STZ compared with sham endothelial cells, showing the down-regulation of *Apold1* and mitochondrial genes, and the up-regulation of interferon-related genes. Dashed lines indicate thresholds for significance: |log$_2$FC| > 0.25 or <−0.25 and −log$_{10}$ (adj *P*-value) ≥ 3. **(C)** Violin plots showing the expression of *Slc38a2* and *Slc38a3* in the different endothelial cell clusters extrapolated by our scRNA-Seq. **(D)** Pathway enrichment analysis of the down-regulated (top) and up-regulated (bottom) genes in enriched endothelial cells from STZ retinas compared with sham retinas from bulk RNA-Seq analysis is shown. Only the most significant differentially expressed genes (log$_2$FC < −1, log$_2$FC > 1 and *P*adj < 0.01) were included in this analysis. **(E)** mRNA expression for *mTOR1*, glutaminolysis (*GLS*, *GLUD1*), autophagy (*ATG5*, *BCLN1*), and apoptosis (*CASP3*, *BAX*, *BCL-2*) genes upon 48 h of high glucose (30 mM) with or without glutamine-deprived (GL-) medium is shown. *Acta2* was used as a housekeeping gene. A summary of three independent experiments is shown. One-way ANOVA followed by Tukey's multiple comparison test was used. *$P$ < 0.05, **$P$ < 0.01, ***$P$ < 0.001, ****$P$ < 0.0001. Normal glucose (n = 3); high glucose (n = 3); mannitol (n = 3), GL- (n = 3), high glucose/GL- (n = 3). Source data are available for this figure.

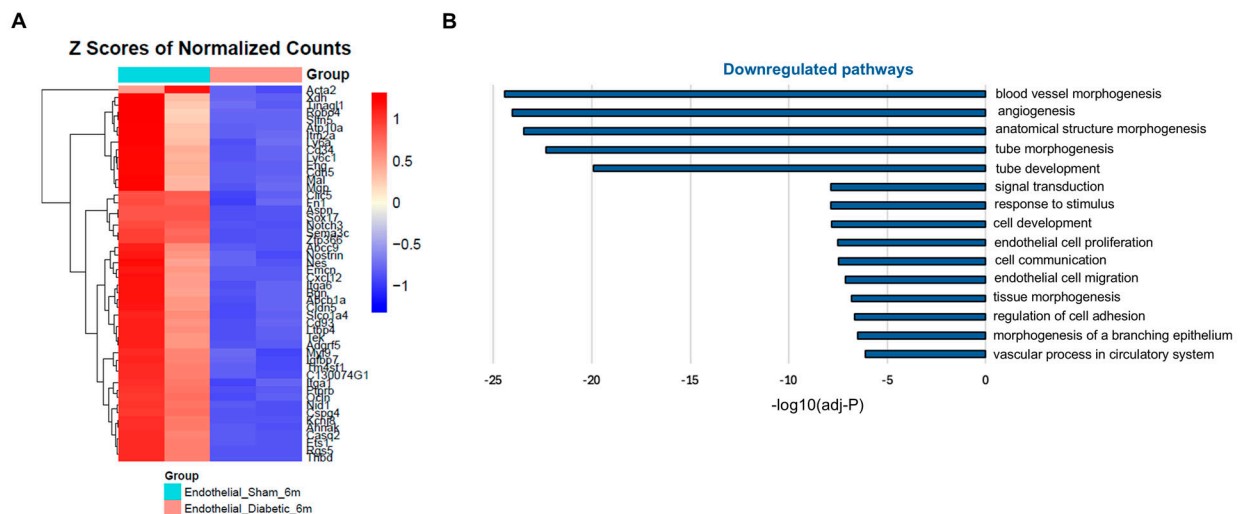

Endothelial cells

**A** Z Scores of Normalized Counts

**B** Downregulated pathways

Retinal cells

**C** Z Scores of Normalized Counts

**D** Upregulated pathways

Downregulated pathways

**E** *P4ah2*    *Col6a1*

**F**

validate this finding, we performed Western blot on whole retinal lysates and found COL6 was expressed in all STZ samples, whereas no band was detected, in the control retinas (Fig 7F). In addition, immunofluorescence of retinal cross-sections at 6 mo showed the expression of COL6 in the diabetic, but not the healthy, retinas (Fig 8A). Moreover, costaining for glutamine synthase (GS), a Müller cell marker, and COL6 showed colocalization of these proteins in diabetic retinas. It is worth mentioning that a significant but small increase in vimentin expression was detected in Müller cells in STZ retinas on scRNA-Seq at the earlier time point of 3 mo, suggesting this process starts at an early stage of disease progression. Cross-sections from diabetic donor eyes confirmed the expression of COL6 in the Müller cell compartment (Fig 8B and C), compared with its absence in control donor eyes, supporting Müller cell contribution to fibrosis in human DR.

# Discussion

In this study, we examined the retinal transcriptional changes in a mouse model of diabetes over the course of 6 mo. Using an integrated approach with bulk RNA-Seq, scRNA-Seq, and in vitro and ex vivo studies, we elucidated the timeline of critical metabolic alterations during DRD progression. Our results highlight the 3-mo time point and Müller cells as particularly critical in the pathophysiology of DRD.

At 3 mo of diabetes, we found significant perturbation of metabolic pathways in Müller cells, endothelial cells, and photoreceptors, suggesting a critical imbalance in the metabolic ecosystem between these cell types. Müller cells play a central role supporting retinal functional and metabolic needs (72), matching the energy supply to the neuronal demands. Müller cells normally rely on lactate released from the photoreceptors (and endothelial cells) to power their TCA cycle (21). The αKG produced by the Müller cell TCA cycle can be then converted to glutamate (via the GDH enzyme, in addition to transaminase reaction) and then to glutamine (GS enzyme), which is made available to other cells for their TCA cycle and metabolic needs (58, 64, 67). In the photoreceptors, glutamine is converted into glutamate, which is either released in synaptic vesicles to stimulate the postsynaptic neurons or shuttled into the mitochondria for energy production. Similarly, the endothelial cells convert Müller cell–derived glutamine into glutamate for mitochondrial TCA cycle and energetic needs (64, 67). Interestingly, our scRNA-Seq, cell culture, and ex vivo retinal culture data suggested reversal of this cycle in the Müller cells,

resulting in a switch to anaplerotic consumption of glutamine for their own metabolic needs and a decreased glutamine availability for the surrounding cells during hyperglycemia. In further support of the alteration of the glutamine–glutamate cycle hypothesis, we also observed a reduction of *Slc1a3* expression in Müller cells, reflecting a reduction in their ability to uptake extracellular glutamate to detoxify the retinal microenvironment. This was corroborated by the accumulation of glutamate in the cell media of primary Müller cells exposed to high glucose in culture and increased glutamate in the diabetic retinal cultured tissue. Literature data have previously shown isolated patch clamp studies of glutamate transporter (SLC1A3 or GLAST) in Müller cells exposed to hyperglycemia, confirming dysfunction of this transporter, with a significant decrease in its activity after 3 mo of STZ-induced diabetes (73). Primary rat Müller cells cultured in high glucose showed significantly decreased glutamate uptake, GS activity, and decreased GLAST and GS expression (74). The perturbation of the glutamine–glutamate cycle impacted the photoreceptors, where we observed down-regulation of *Slc38a3*, the amino acid transporter specific for shuttling glutamine from the extracellular space into the rod and cone photoreceptors. This reduction was confirmed at the protein level, indicating decreased capacity to uptake the glutamine necessary to produce the neurotransmitter glutamate and to fuel the TCA cycle and photoreceptor metabolism and survival. More specifically, glutamine plays a critical role in rod photoreceptors by supporting the production of key metabolites, including aspartate, which are essential for metabolic homeostasis and protein synthesis (75). Rod-specific *Gls* knockout (KO) studies have demonstrated that disruption of glutamine metabolism triggers a stress response that leads to severe photoreceptor degeneration and functional loss (75), highlighting the indispensable role of glutamine metabolism in maintaining photoreceptor health. Unlike alteration in glucose or other metabolic pathways, which often activate adaptive mechanisms and redundancies to preserve metabolic balance, impairments in glutamine metabolism cannot be compensated for, resulting in rapid photoreceptor degeneration (75, 76). Another metabolite derived from glutamine, glutamate, is the most important excitatory neurotransmitter of the ribbon synapse of the photoreceptor axonal terminals (77), specialized synapses crucial for the rapid transmission of visual information from the retina to the brain. To validate the downstream effect of Müller cell anaplerosis on the ribbon synapse, we confirmed significant down-regulation of photoreceptor *Cacnb2*, that encodes an important Ca$^{2+}$-voltage channel, which regulates the exocytosis of neurotransmitter vesicles at the ribbon synaptic terminal (78). Extracardiac, global

**Figure 7. Endothelial and retinal gene expression at 6 mo from bulk RNA-Seq.**
**(A)** Heatmap of the top 50 up-regulated (red) and down-regulated (blue) genes comparing endothelial cells isolated from sham (cadet blue) and STZ (red) retinas. **(B)** Pathway enrichment analysis of the down-regulated genes in enriched endothelial cells from STZ retinas compared with endothelial cells isolated from sham retinas is shown. Only the most significant differentially expressed genes (log$_2$FC < −1 and *P*adj < 0.01) were chosen for the analysis. **(C)** Heatmap of the top 50 up-regulated (red) and down-regulated (blue) genes comparing retinal cells isolated from sham (cadet blue) and STZ (red) retinas. **(D)** Pathway enrichment analysis of the up-regulated (top) and down-regulated (bottom) genes in retinal cells from STZ retinas compared with sham retinas is shown. Only the most significant differentially expressed genes (log$_2$FC > 1 or <1 and *P*adj < 0.01) were chosen for the analysis. **(E, F)** mRNA expression of profibrotic markers *P4ah2* and *Col6a1* from 6-mo sham and STZ retinas (*Acta2* was used as a housekeeping gene) and (F) Western blot analysis for COL6 in 6-mo sham and STZ retina protein lysates. Representative bands are shown. Band densitometric analysis results are shown in the graphs on the right. β-Actin (BACT) was used as a loading control. The Mann–Whitney *U* test was used for statistical analysis. *P < 0.05. sham (n = 4); STZ (n = 4).
Source data are available for this figure.

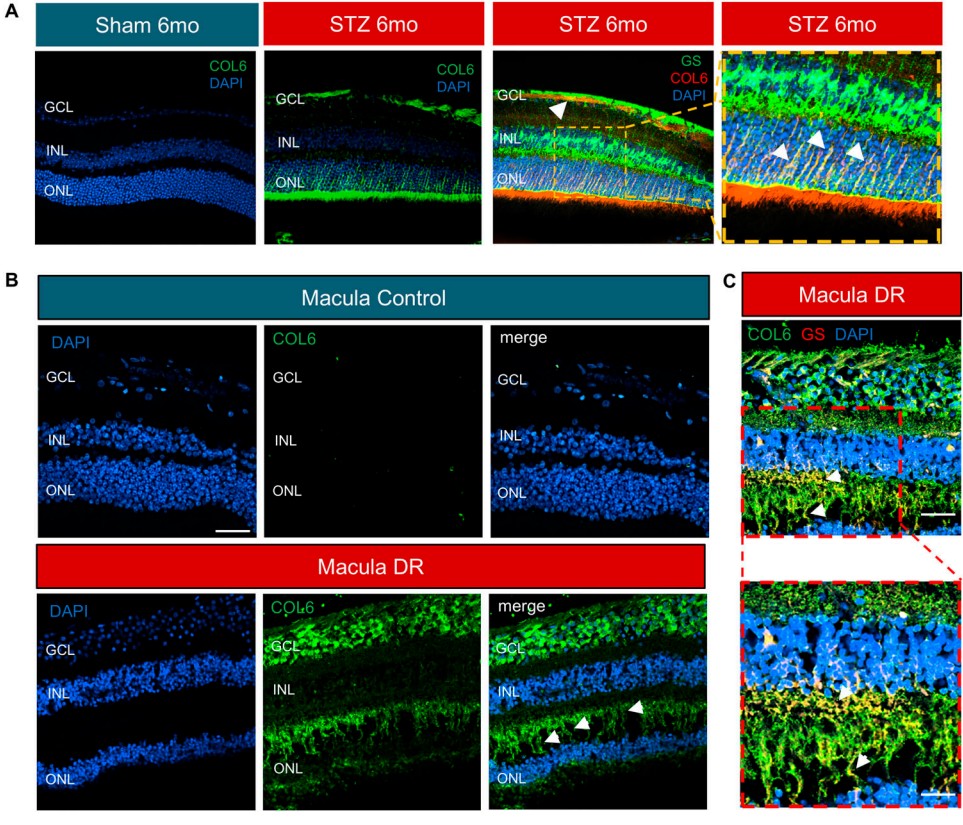

**Figure 8. Fibrosis quantification in mouse and human diabetic samples.**
**(A)** COL6 staining on paraffin-embedded histological cross-sections of 6-mo STZ murine eyes compared with sham eyes. GS labeling of Müller cells costained with COL6 on paraffin-embedded histological cross-sections of 6-mo STZ mouse retina is shown on the right, with an insert showing higher magnification of an area of interest. Arrowheads indicate the COL6+/GS+ cells in the retina. DAPI was used to counterstain the nuclei. The retinal layers are labeled. Scale bar: 20 $\mu$m, 50 $\mu$m. **(B)** Representative images of COL6 staining on paraffin-embedded histological cross-sections of the macula of human donor with diabetic retinopathy (bottom) compared with control donor (top) are shown. **(C)** GS labeling and costaining with COL6 on paraffin-embedded histological cross-sections of the DR macula, with an insert showing a higher magnification of an area of interest. Arrowheads indicate the COL6+/GS+ cells in the human retina. DAPI was used to counterstain the nuclei. The retinal layers are labeled. Scale bar: 20 $\mu$m, 50 $\mu$m.

deletion of *Cacnb2* has been shown to alter retinal synaptic morphology, RIBEYE and PSD95 distribution and expression, and ribbon synaptic transmission (79). Extracardiac, global deletion of *Cacnb2* has been shown to alter retinal synaptic morphology, RIBEYE and PSD95 distribution and expression, and ribbon synaptic transmission (79). Interestingly, *CACNB2* has also been described as a novel susceptibility gene in DRD, where variants of the *CACNB2* gene were associated with proliferative DR in genome-wide analyses (80). Our data showed altered distribution of RIBEYE and PSD95 in diabetic mouse retinas, and significant reduction of both proteins in mouse retinal lysates. We further validated these changes in human donor eye specimens, where we observed significant reduction and disorganization of PSD95 and RIBEYE expression in the diabetic macula. We would expect that these synaptic structural alterations, together with the impaired glutamine–glutamate cycle, contribute to altered photoreceptor synaptic signal transmission in diabetes. This is in line with human ERG studies that have provided convincing evidence for progressive photoreceptor dysfunction in diabetes (81, 82, 83, 84), where the scotopic (rod-mediated) oscillatory potential is delayed in early stages of the disease, and the photopic delay has been observed in established DR, suggesting that cones may be involved in later stages (84). Photoreceptor dysfunction is also corroborated in rodent models of diabetes (85, 86, 87).

In view of the anaplerotic switch of the Müller cells, we analyzed the metabolic perturbation in endothelial cells in our scRNA-Seq data, considering their reliance on Müller cell–derived glutamine to fuel their TCA cycle. Blocking the glutamine pathway has been shown to cause a collapse of TCA cycle intermediates in endothelial cells, disrupting their ability to proliferate (67, 88). In addition, Singh et al showed that hyperoxia leads to glutamine-fueled anaplerotic switch in Müller cells, with reduction in glycolytic entry into the TCA cycle, and reversal of Müller metabolism from production to consumption of glutamine (64). In response to this metabolic switch, endothelial cells enhance their glutaminolysis, increasing the extracellular toxic ammonium, leading to arrest of endothelial cell growth (64, 67, 89). Our scRNA-Seq data did not show significant alteration in the endothelial expression of glutamine transporter (*Slc38a2*, *Slac38a3*) and glutaminolysis (*Gls*). We suspect that this might be related to disproportionate loss of dysfunctional endothelial cells during single-cell preparation, as these cells may be more vulnerable. Our cultured endothelial cell experiments showed that in response to glutamine deprivation, whether in the setting of high or normal glucose, endothelial cells downregulated the expression of glutamate and $\alpha$KG-producing enzymes and showed increased expression of pro-apoptotic genes. Intriguingly, the combination of high glucose and glutamine starvation did not appear to be additive; our data show that high glucose alone without glutamine deprivation did not influence endothelial glutamine metabolism or apoptosis, whereas glutamine deprivation alone was detrimental for these pathways. We are aware that an in vitro systems cannot fully recapitulate the complexity of the in vivo environment, but our aim was to validate a specific aspect of endothelial

cell behavior, and our data confirm that glutamine deprivation, likely as a consequence of Müller cell anaplerosis and downstream metabolic deregulation, may be a critical cause of endothelial cell dysfunction in the setting of hyperglycemia, though other factors might additively and cumulatively culminate in endothelial cell death at 6 mo in the STZ model.

Endothelial cell activation and inflammation are recognized as early events in DR. Increased expression and secretion of chemokines and cytokines from endothelial cells contribute to immune cell adhesion and vascular permeability (90, 91), further damaging the endothelial cells. Our bulk RNA-Seq showed an endothelial inflammatory state that started as early as 1 mo and continued at 3 mo, with additional alteration of molecules involved in metabolism and an increased vascular leakage in the deep layer, confirming a progressive loss of vascular stability in diabetic retinas. The decline of endothelial cells and vascular leakage in the deep plexus can also be linked to synaptic glutamate alterations observed at 3 mo. Previous studies have shown that glutamatergic neuronal activities regulate vascular development and BRB maturation, a phenomenon that could be relevant for retinal disease as well (92). In *Vglut−/−* mice, delayed deep plexus angiogenesis and paracellular vascular leakage were observed when compared with WT mice (92); this aligns with our findings of increased biocytin leakage in STZ retinas compared with sham, coinciding with dysregulation of the glutamine–glutamate cycle and impaired synaptic morphology. More interestingly, the same study (92) showed that GS and GLAST are down-regulated in the *Vglut−/−* mice and that GLAST activity mediates the induction of pro-angiogenic pathways such as Norrin and WNT. Notably, we observed a significant decrease in the WNT pathway as early as 1 mo in our study, which together with glutamine–glutamate cycle dysregulation may contribute to cumulative vascular damage over time.

The most dramatic endothelial defects were seen at 6 mo of diabetes, with down-regulation of angiogenesis and vasculogenesis pathways in bulk RNA-Seq, and a reduction in pericyte coverage in STZ retinal blood vessels, indicating generalized functional decline of the endothelial cells. This fits with the timeline of vascular defects observed in the STZ model, such as acellular capillaries, vascular cell apoptosis, and pericyte loss, generally seen at 6 mo (93), and progressing at 9 mo (26). We also observed significant up-regulation of fibrotic pathways and activation of collagen synthesis machinery in the retinal compartment at 6 mo, with up-regulation of *Col6a1* and *P4ah2* at the RNA level. This was corroborated by increased COL6 in retinal protein lysates, and immunostaining evidence of collagen expression in the Müller cells and beyond in the diabetic retina. Importantly, we also confirmed COL6 colabeling of the Müller cells in the macula of diabetic human donor eyes. In the human retina, fibrotic scarring may compromise vision and ultimately lead to blindness, especially in proliferative DR, where fibrovascular preretinal scar tissue contracts to elevate the retina, detaching it from the underlying RPE layer, and causing vision-threatening retinal detachment (94, 95, 96). Diverse cell types can contribute to the development of fibrotic tissue such as RPE cells, inflammatory cells, endothelial cells, and pericytes (97, 98, 99). In addition, Müller cells, together with astrocytes, contribute to retinal gliosis (100, 101). Our data indicate that although the STZ mouse model does not progress to preretinal fibrovascular membranes, as seen in advanced proliferative stages of the human disease, the triggers of fibrogenesis and

pathways regulating the production and assembly of extracellular matrix proteins are in fact activated in the diabetic mouse.

The strength of our study lies in the use of an unbiased, integrative approach to elucidate the timeline and molecular mechanisms involved in DRD, with validation in human donor retinal tissue. Complementary information from bulk RNA-Seq and scRNA-Seq allowed us to study endothelial cell dysfunction, and enabled us to identify the 3-mo time point as crucial, with Müller cells as central drivers of metabolic dysfunction in the photoreceptors, ribbon synapses, and endothelial cells, linking neurodegeneration and vascular degeneration and visual dysfunction in human DRD.

Limitations are related to the processes for generating the single-cell suspension, which limited the recovery of inner retinal neurons: ganglion, amacrine and horizontal cells, and pericytes and immune cells (102, 103, 104). As well, these harsh processes substantially limited endothelial cell (despite enrichment). Further study will be necessary to optimize the digestion protocol and to pair scRNA-Seq with single-nucleus RNA sequencing, which has been more effective at recovering these cell types (105, 106).

In conclusion, our study revealed Müller cell (macroglial) metabolic dysregulation and glutamine starvation impact endothelial and neuronal degeneration, linking diabetic vascular and neural degeneration in mice and humans, although further studies are needed. These findings suggest that addressing the glutamate–glutamine cycle imbalance may mitigate the neuronal and endothelial cell dysfunction in diabetes, and targeted metabolic interventions are needed to ameliorate Müller cell dysfunction and thereby prevent or delay the onset of DRD and its sight-threatening complications.

# Materials and Methods

## Animals

The experimental protocols in this study were approved by the Institutional Animal Care and Use Committee at Northwestern University and were conducted in accordance with the *Guide for the Care and Use of Laboratory Animals of the National Institutes of Health*. A mixture of male and female on a mixed background of C57BL/6J and FVB (The Jackson Laboratory) was used for all animal experiments in this study. Mice were genotyped for the absence of the Crb1[rd8] mutation.

## Human donor eyes and study approval

All human samples were received from the Eversight Eye Bank, a process that was exempt from IRB approvals in our department. All the procedures were in accordance with Declaration of Helsinki for research involving human subjects. Detailed information about the donors is reported in Table 1.

## STZ-induced diabetes

Pharmaceutical-grade streptozotocin (STZ; Sigma-Aldrich) was dissolved in citrate buffer (0.04 M sodium citrate, 0.06 M citric acid, pH 4.0). Animals aged 7–8 wk were fasted for 6 h before STZ

**Table 1. Detailed clinical information of human eye donors used in the study.**

| Sample | Age/sex/race | Type/duration of diabetes | Cause of death |
|---|---|---|---|
| C1 | 71/M/NA | NA | COPD |
| C2 | 72/F/NA | NA | Brain cancer |
| C3 | 74/F/NA | NA | COPD |
| C4 | 83/F/White, non-Hispanic | NA | Unknown |
| C5 | 84/M/White, non-Hispanic | NA | Prostate cancer with Mets |
| C6 | 84/M/White, non-Hispanic | NA | COPD |
| D1 | 57/M/Caucasian | Type 2/15-25 yr | Respiratory failure |
| D2 | 62/M/Caucasian | Type 2/10-15 yr | End-stage renal disease |
| D3 | 35/M/Asian | Type 1/2 PDR/12 yr | Pulmonary embolism |
| D4 | 97/M/White, non-Hispanic | Type 2/NA | PVD, Gangrene |
| D5 | 81/M/White, non-Hispanic | Type 1/NA | Colon cancer with Mets |
| D6 | 67/M/White, non-Hispanic | Type I/NA | Cerebral vascular accident |

C, control; D, diabetes; M, male; F, female; NA, not available; PDR, proliferative diabetic retinopathy; COPD, chronic obstructive pulmonary disease; PVD, peripheral vascular disease.

administration, and base line weight and blood glucose levels were taken. A single-dose intraperitoneal injection of 150 mg STZ per kg body weight for females and 100 mg/kg for males was administered once. An equivalent dose of citrate buffer was given for age-matched control animals. To prevent hypoglycemic shock, animals were provided with 10% sucrose water for the first night after STZ injection. Five days after STZ administration, animals with blood sugars greater than 300 mg/dl were considered diabetic. Animals were weighed weekly to monitor diabetes-associated weight loss, and blood sugars were recorded monthly until 1, 3, and 6 mo of diabetic experimental endpoints were reached. Blood glucose measurements were taken at random time points, as the mice had ad libitum access to food. If an animal lost more than 15% of their peak body weight, 2 U/kg of insulin (Hanna Pharmaceuticals) was administered twice per week.

### Collection and dissociation of diabetic retinas

After euthanasia, enucleated globes were placed in PBS (Invitrogen) and stored on ice for transportation. For dissociation, dissected retinas were transferred to a 500 $\mu$l solution of 0.22-$\mu$m filter-sterilized 1x Dulbecco's phosphate-buffered saline (DPBS) with calcium and magnesium (Life Technologies) containing 2 mg/ml collagenase type I (Worthington). Samples were incubated at 37°C for 30 min with gentle trituration every 10 min. Samples were then spun down at 400$g$ for 8 min at 4°C, and cell pellets were resuspended in 500 $\mu$l 1X DPBS without calcium and magnesium (Life Technologies), 1% bovine serum albumen (BSA; Sigma-Aldrich), and 2 mM EDTA and stored on ice until further processing.

### Endothelial cell isolation and RNA purification

A rat anti-mouse CD31 antibody (Invitrogen) was bound to sheep anti-rat Dynabeads (Invitrogen) following the manufacturer's instructions. 10 $\mu$l of the Dynabeads complex was added to the single-cell suspension and incubated at 4°C for 1 h with gentle rotation. Cells bound to the CD31–Dynabead complex were pulled down with a magnet, and supernatants containing unbound cells were collected. Cell supernatants were pelleted at 400$g$ for 8 min at 4°C. CD31-enriched cell pellets were washed three times with 1x DPBS. All cell pellets were lysed in RNA Stat-60 solution (Tel Test B Labs) overnight at –80°C. Nucleic acids were isolated with chloroform, precipitated with 100% ethanol, and washed with 70% ethanol. DNase reaction was performed following the manufacturer's suggested protocol (Thermo Fisher Scientific). RNA was extracted with phenol/chloroform (Ambion) and precipitated overnight in 100% ethanol/160 mM ammonium acetate/0.8% glycogen at –80°C. RNA pellets were resuspended in 10 $\mu$l nuclease-free ddH$_2$O.

### RNA sequencing (bulk RNA-Seq)

All bulk RNA-Seq experiments were performed at the Functional Genomics Core at the University of Chicago (Chicago, IL). RNA quality and quantity were assessed using an Agilent Bioanalyzer. RNA-SEQ libraries were prepared using Illumina mRNA TruSeq kits following the manufacturer's instructions (Illumina). Library quality and quantity were checked using an Agilent Bioanalyzer (Agilent), and the pool of libraries was sequenced using an Illumina HiSeq 4000 (single-end 50 bp) using Illumina reagents and protocols.

### Differential expression analysis

Reads were trimmed to remove Illumina adapters from the 3′ ends using cutadapt ([107]). Trimmed reads were aligned to the *Mus musculus* genome (mm10) using STAR ([108]). Read counts for each gene were calculated using htseq-count ([109]) in conjunction with a gene annotation file for mm10 obtained from Ensembl (http://useast.ensembl.org/index.html). Normalization and differential

**Table 2.  Primer sequences used for qRT-PCR.**

| Target | Forward | Reverse |
|--------|---------|---------|
| H2Aa | GAACCTGAGATTCCAGCCCC | GAAGAGGGACACACGCCTTC |
| H2Ab1 | GATCAAAGTGCGCTGGTTCC | CAGACTGTGCCCTCCACTC |
| Gbp6 | AGAGCAGAAGGGAGCCTGAG | GGGTCATAGCAACCTGGTCC |
| Iigp1 | ACAGGAGTTTCTGATGCACAC | CACTCACTGCTCCTGAGAGG |
| Rho | GCCCACAGGATGCAATTTGG | CTCGGGGATGTACCTGGAC |
| Rdh8 | TGCTTCCAGGCATGAAGAGG | CGTTGAAGATGACACCCTGC |
| P4ah2 | CTATTCATCCATGGACCGGCA | TGGGAGTACGGGTCTTGACT |
| Col6a1 | CGATTGCCTTCCAAGACTGC | AGGCCCCATAAGGTTTCAGC |
| Acta2 | CTGTCGAGTCGCGTCCACC | TCGTCATCCATGGCGAACTGG |
| GLUL | AGAGTGGGAGAAGAGCGGAG | TCATGGTGGAAGGTGTTCTGG |
| GLS | GGGAATTCACTTTTGTCACGATCT | AAGGAATGCCTTTGATCACCAC |
| GLUD1 | TCTAGGAAAGGGAATGACCTGC | CCCAAACGGCACATCAACC |
| SLC3A1 | CCCTTACAAAATCAGAAAAGTTGTG | CCCCATCTTGGGCTCTTCTC |
| RLBP1 | AGTCACAACTTGGCCCTGAC | TTCAGGTACCATGCGGAACG |
| mTOR1 | CTCCCGGCTTAGAGGACAGC | AGTTTCCTTTAATATTCGCGCGGC |
| ATG5 | CGGGTGAAGGTGGTTCCT | TTTCAACCAAAGCCAAACTTAGTA |
| BCLN1 | TCCAGGAACTCACAGCTCCAT | TCTGCGAGAGACACCATCCT |
| CASP3 | AGGCGGTTGTAGAAGAGTTTCG | GCTCGCTAACTCCTCACGG |
| BAX | GATGGACGGGTCCGGGG | TCGATCCTGGATGAAACCCTG |
| BCL-2 | GAGTTCGGTGGGGTCATGTG | CAGTTCCACAAAGGCATCCCAG |
| BACT | GTCATTCCAAATATGAGATGCGT | TGTGGACTTGGGAGAGGACT |

expression analysis were performed using R v3.6.0 with the Bioconductor-DESeq2 package 1.14.1 that employs the Wald test (110). Results were corrected for multiple testing using the Benjamin–Hochberg method and reported as FDR-adjusted P-value. Significance in DESeq2 was determined just by an FDR-adjusted P-value less than 0.05. Heatmaps of DEGs were prepared using R.

### Pathway enrichment analysis

The functional enrichment analysis was performed using Metascape (https://metascape.org) (111) to identify significant pathways among the significantly differently expressed genes. GO processes, KEGG pathways, Reactome gene sets, canonical pathways, and CORUM complexes were used as sources for the analysis.

### Retinal digestion for single-cell RNA sequencing (scRNA-Seq)

Retinas were dissected in ice-cold 1XPBS and placed in 1 ml of 10% FBS/DMEM solution containing 0.5 mg/ml collagenase A, 0.5 mg/ml collagenase I, 1 $\mu$M Rock inhibitor Y27362 (ACS-3030; ATCC), and 2x endothelial cell growth serum (PromoCell). The retinas were incubated for 25 min at 37°C with shaking at 6$g$. Retinas were pipetted to dissociate cells at 10-, 20-, and 25-min time points, strained using 40-$\mu$m nylon mesh screen, and washed with 2%

FBS/HBSS. Cells were treated with a dead cell removal kit (130-090-101; Miltenyi Biotec) to remove dead cells and enriched for endothelial cells using CD31 microbeads and MS column (Miltenyi Biotec). CD31-negative cell fractions were incubated with Dynabeads (Invitrogen) treated with anti-CD73 clone TY/11.8 (Becton Dickinson) to negatively select photoreceptor cells. CD73-negative unbound cells were collected and mixed with the original CD31-positive enriched cell fraction in 5:1 ratio totaling ≈10,000 cells both for sham and for STZ sample. The cell suspension was submitted to single-cell RNA-Seq analysis. Single-cell libraries were generated using a Chromium Single Cell 3′ v3 kit (10x Genomics), processed, and analyzed as previously described (99).

### Quantitative reverse transcription PCR (qRT-PCR)

Total RNA was extracted as described above. 1 $\mu$g of RNA, quantified with a NanoDrop, was reverse-transcribed with the SuperScript III First-Strand Synthesis System for RT–PCR kit (18080-051; Thermo Fisher Scientific) and subjected to quantitative PCR using the PowerUp SYBR Green Master Mix (A25742, Applied Biosystems by Thermo Fisher Scientific) with the QuantStudio 3 Real-Time System (Applied Biosystems by Thermo Fisher Scientific). Real-time data were collected for 40 cycles of 95°C, 15 s; 60°C, 60 s. Primers used are custom-synthesized by Integrated DNA Technology, and their sequence is reported in Table 2. The relative expression of the genes of interest was evaluated using the ΔΔCt method with beta-actin (Acta2) as a reference gene.

### Histological preparation

Enucleated globes were fixed in 4% PFA overnight at 4°C and then washed in PBS for histological preparation. Human globes were fixed in 10% normal formalin buffer (NFB). After fixation, eye paraffin embedding was conducted by the Mouse Histology and Phenotyping Core Laboratory of Northwestern University. Samples were then sectioned (7 $\mu$m) for further analysis. For cryosection preparation, after fixation the globes were washed in PBS, then left in a 30% sucrose solution overnight (ON) at 4°C, before being embedded in optimal cutting temperature compound (OCT) and sectioned using a cryostat.

### Immunofluorescence, imaging, and analyses

Paraffin retina cross-sections were deparaffinized and rehydrated, antigens were retrieved by steaming with sodium citrate, pH 6.0, at 90°C for 20′, and then endogenous peroxidase blocking was performed in MetOH/3%$H_2O_2$ for 10′ at RT. All the sections (paraffin and cryosections) underwent nonspecific binding blocking step with 1XPBS/1%BSA/5%DS/0.3% Triton X-100 for 1 h at RT. Sections were incubated with primary antibodies overnight at 4°C, followed by incubation with secondary antibodies for 2 h at RT. The cross-sections were then stained with 0.5% Sudan black to dampen down photoreceptor autofluorescence. Then, 4′,6-diamidino-2-phenyl-indole counterstaining was implemented to visualize the nuclei (DAPI; Thermo Fisher Scientific). The slides were then mounted with ProLong Gold Antifade Reagent (Thermo Fisher Scientific). For retina flat mount staining to analyze the retinal vascular

**Table 3.** List of antibodies used for immunofluorescence and Western blot.

| Target | Source | Company | Catalog # | Dilution | Detection |
|---|---|---|---|---|---|
| PSD95 | Guinea pig | Synaptic Systems | 124014 | 1:500 | IF |
| RIBEYE | Mouse | BD | 612044 | 1:1,000 | IF |
| SLC38A3 | Rabbit | Thermo Fisher Scientific | 14315-1-AP | 1:50–1:1,000 | IF-WB |
| EAAT5 | Rabbit | Sigma-Aldrich | HPA049124 | 1:500 | IF |
| GS | Mouse | Abcam | ab64613 | 1:75–1:1,000 | IF-WB |
| GLAST | Rabbit | Osenses | OSE00018G | 1:200–1:1,000 | IF-WB |
| COL6 | Rabbit | Abcam | ab182744 | 1:50–1:1,000 | IF-WB |
| NG2 | Rabbit | Sigma-Aldrich | ZRB5320 | 1:200 | IF |
| CD31 | Rat | Becton Dickenson | 550274 | 1:100 | IF |
| COL4 | Goat | Rockland | 600101MN4 | 1:250 | IF |
| CRALBP | Mouse | Thermo Fisher Scientific | MA1-813 | 1:200 | IF |
| BACT | Rabbit | Abcam | Ab8227 | 1:5,000 | WB |
| Secondary Ab | Source | Company | Catalog # | Dilution | Conjugate |
| Guinea pig | Donkey | Jackson ImmunoResearch | 706-605-148 | 1:200 | Alexa Fluor 647 |
| Mouse | Donkey | Jackson ImmunoResearch | 715-545-150 | 1:200 | Alexa Fluor 488 |
| Rabbit | Donkey | Jackson ImmunoResearch | 711-545-152 | 1:200 | Alexa Fluor 488 |
| Mouse | Donkey | Jackson ImmunoResearch | 715-025-151 | 1:200 | Alexa Fluor 568 |
| Rat | Donkey | Jackson ImmunoResearch | 712-025-153 | 1:200 | Alexa Fluor 568 |
| Rabbit | Donkey | Jackson ImmunoResearch | 711-605-152 | 1:200 | Alexa Fluor 647 |
| Goat | Donkey | Jackson ImmunoResearch | 705-545-147 | 1:200 | Alexa Fluor 488 |
| Rabbit | Goat | Thermo Fisher Scientific | 32460 | 1:10,000 | HRP |
| Mouse | Goat | Thermo Fisher Scientific | 31430 | 1:10,000 | HRP |

compromise at 6 mo, retinas were dissected after fixation and treated with 100% methanol for 2 hrs at –20°C, before undergoing nonspecific binding blocking step. Retinas were incubated with primary antibodies for 4 d at 4°C, followed by incubation with secondary antibodies overnight at 4°C.

The Nikon W1 Dual Cam spinning disk confocal laser microscope system (Nikon) and the Nikon SoRa spinning disk microscope system at the Center for Advanced Microscopy/Nikon Imaging Center of Northwestern University were used to image all the staining. NIH ImageJ software (ImageJ, 1.54d, National Institutes of Health) was used for image analyses. For the quantification of corrected total fluorescence and/or staining area of PSD95, RIBEYE, and GS, three retina fields in each animal were analyzed at a 20x magnification.

To evaluate retinal vascular compromise at 6 mo, ghost vessel length was quantified using ImageJ. Two retinal fields from each animal were imaged at 40x magnification, including the three capillary layers. Ghost vessels were defined as capillary segments that are COL4+ and CD31⁻, indicating de-endothelialized capillary tubes. These segments were quantified in the z-stacks and analyzed in each capillary layer. That data are presented for the combined superficial, intermediate, and deep capillary layers, as well as the deep layer alone. For NG2+ cell/mm of vessel quantification, four retinal fields from each animal were imaged at 40x magnification in the central

part of the retina, including the three capillary layers. Analyses were performed blinded and averaged for each animal. Data are shown for the combined superficial, intermediate, and deep capillary layers.

## Western blot

Retinal tissues were lysed in RIPA buffer (R0278; Sigma-Aldrich) supplemented with protease and phosphatase inhibitors (11836170001 and 04906845001; Roche) for 20 min on ice. The samples were then homogenized using the TissueLyser LT (QIA-GEN) for 10 min at 4°C and cleared by centrifugation. The Bradford assay (23238; Thermo Fisher Scientific) was used to measure protein concentration. A total of 30 µg proteins were denatured and separated on a 3–8% Tris-acetate gel (EA0375BOX; Invitrogen) or 4–12% Bis-Tris gel (NP0321BOX; Invitrogen), followed by transfer onto a nitrocellulose membrane. Membranes were blocked in 5% milk (or 5% BSA) at RT for 1 h, then incubated with primary antibodies overnight at 4°C. Membranes were washed in 0.1% TBS-T and then incubated with the HRP-conjugated antibodies for 2 h at RT. Immunoreactive bands were visualized with a chemiluminescent substrate (34577; Thermo Fisher Scientific), and the iBright1500 imager (Invitrogen) was used to acquire the images. See Table 3 for primary and secondary antibodies used.

**MIO-M1 cell culture and high-glucose treatment**

The human Müller cell line Moorfields/Institute of Ophthalmology-Müller 1 (MIO-M1) was obtained from the UCL Institute of Ophthalmology, London, UK ([112]). MIO-M1 cells were cultured in low-glucose DMEM (1g/liter; 5 mM, 11885-084; Gibco), containing pyruvate (110 mg/liter) and glutamine, supplemented with 10% FBS and antibiotics (1% P/S). For the high-glucose experiments, the cells were then cultured in 30 mM D-glucose or 24.6 mM mannitol for 24 and 48 h, without replacing the culture media after the first 24 h of culture. After the time course, cell media were collected, and proteins and RNA were extracted for molecular analyses. The mannitol concentration (24.6 mM) corresponds to the difference between the high-glucose (30 mM) and low-glucose (5.5 mM) media, serving as an osmotic control to account for the increased osmolarity in the high-glucose condition.

**Glutamine and glutamate quantification assay**

After the time course and high-glucose treatment, a colorimetric assay to quantify intracellular and extracellular glutamine (ab197011; Abcam) and glutamate (ab83389; Abcam) was used following the manufacturer's instructions. The colorimetric product of the reactions was read at 450 nm using a plate reader, and the cell number was used to normalize the intracellular glutamine and intracellular glutamate amount detected.

**Ex vivo retinal culture, metabolite extraction, and mass spectrometry**

3-mo sham and STZ retinas were harvested and cultured in Krebs–Ringer–Hepes/bicarbonate medium supplemented with 5 mM glucose, 5 mM aspartate, and 5 mM lactate for 90' as previously reported ([58], [66]). After the culture, the extraction of hydrophilic metabolites (MetOH based) was performed on the retinal tissue and culture media. Briefly, 80% cold MetOH was added to the tissue (or media) that was subsequently homogenized. Samples were then incubated at –20°C overnight in 80% MetOH. The next day, samples were vortexed for 30" and spun down at 20,000$g$ for 20'. The supernatant was submitted to the Metabolomics Core for Targeted Metabolites mass spectrometry. Total protein amount (for retinal tissue samples) or initial volume (for culture media) was used to calculate the mass spectrometry injection. Raw peak areas were normalized to total ion count and lactate levels.

**Human retinal microvascular endothelial cell culture and treatments**

Human retinal microvascular endothelial cells (HRMEC) (ACBRI 181; Cell Systems) were cultured in EBM-2–containing EGM supplements (CC-3156 and CC-4133; Lonza). For high-glucose and glutamine deprivation experiments, DMEM with or without glutamine (11885-084 and 11054-020, respectively; Thermo Fisher Scientific) was used with supplementation of EGM bullet kit (CC4133; Lonza).

Cells were cultured in mM D-glucose or 24.6 mM mannitol for 48 h, and then, the RNA was extracted for downstream molecular analyses.

**Leukostasis (cardiac perfusion assay)**

45-d STZ-injected mice (and sham-injected controls) were anesthetized with ketamine: xylazine cocktail (100:20 mg/kg) administered via intraperitoneal (IP) injection. A surgical lateral incision of ~5–6 cm was made through the abdominal wall, and the rib cage was cut to expose the pleural cavity. A 25 G 5/8 needle was then inserted into the left ventricle, and 5 ml of cold PBS was injected followed by 3 ml of FITC-Concanavalin (Vector Labs)/PBS at 1:15 dilution. An additional 5 ml of cold PBS was injected. The eyes were then enucleated and fixed in 4% PFA/PBS, and washed, and retinas were dissected and mounted. The leukocytes were counted in the whole retinas to obtain an accurate count using a Nikon 80i Eclipse microscope at a 40X magnification (Nikon).

**Biocytin assay**

3-mo STZ-injected mice (and sham-injected controls) were injected with 100 $\mu$l of 1% biocytin TMR (Invitrogen) into the tail vein. Mouse eyes were enucleated after 30 min and processed for staining. Central retinas were imaged and scored for the number of positive biocytin TMR in the vascular layers and quantified.

**Statistical analysis**

Statistical analyses were performed using GraphPad Prism version 10 software (GraphPad Software) for the two-tailed Mann–Whitney test, Welch's $t$ test, or one-way ANOVA. The data are reported as the mean ± SEM. Statistical significance was defined as $P < 0.05$.

# Data Availability

Bulk RNA-Seq and single cell RNA-Seq datasets associated with this study have been uploaded to GEO with accession GSE313176.

# Supplementary Information

# Acknowledgements

Imaging work was performed at the Northwestern University Center for Advanced Microscopy (RRID: SCR_020996) generously supported by NCI CCSG P30 CA060553 awarded to the Robert H Lurie Comprehensive Cancer Center. Super-resolution spinning disk microscopy was performed on a Nikon SoRa system, purchased through the support of NIH 1S10OD032270-01. Metabolomics experiments were performed by Metabolomics Core Facility at Robert H. Lurie Comprehensive Cancer Center (generously supported by NCI CCSG P30 CA060553) of Northwestern University. This work was supported by the Northwestern University NUSeq Core Facility. We

thank Dr. Matthew Schipma (Northwestern University, NUSeq), Dr. Steven H DeVries for helpful discussion about the article, Dr. Robert F Mullins for sharing human retinal specimens, and Dr. Astrid Limb (UCL Institute of Ophthalmology, London, UK) for kindly providing the MIO-M1 cell line. The present work was supported by the following grants: NIH-R01-EY30121-A1 (to AA Fawzi).

## Author Contributions

K Corano Scheri: conceptualization, formal analysis, investigation, visualization, and writing—original draft.
Y-W Hsieh: investigation.
T Tedeschi: investigation.
JB Hurley: conceptualization and writing—review and editing.
AA Fawzi: conceptualization, supervision, funding acquisition, visualization, and writing—review and editing.

## Conflict of Interest Statement

AA Fawzi is a consultant to Regeneron, Roche/Genentech, Boehringer Ingelheim, and RegenXbio, but these entities did not have any relevant role in this article. T Tedeschi is currently employed at Regeneron.

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
