## [Reviewer comments · Life Science Alliance]

Müller cell glutamine metabolism links photoreceptor and endothelial injury in diabetic retinopathy

Katia Corano Scheri, Yi-Wen Hsieh, Thomas Tedeschi, James Hurley, and Amani Fawzi
DOI: <https://doi.org/10.26508/lsa.202503434>

Corresponding author(s): Amani Fawzi, Northwestern University

Review Timeline:

Submission Date:	2025-06-25
Editorial Decision:	2025-08-07
Revision Received:	2025-10-10
Editorial Decision:	2025-11-03
Revision Received:	2025-11-05
Accepted:	2025-11-11

Scientific Editor: Tim Fessenden

Transaction Report:

August 7, 2025

Re: Life Science Alliance manuscript #LSA-2025-03434-T

Dr. Amani A. Fawzi
Northwestern University
Ophthalmology
645 N. Michigan Ave # 440
Chicago, IL 60611

Dear Dr. Fawzi,

Thank you for submitting your manuscript entitled "Müller cell glutamine metabolism link photoreceptor and endothelial injury in diabetic retinopathy" to Life Science Alliance. The manuscript was assessed by expert reviewers, whose comments are appended to this letter.

As you will see, reviewers overall appreciated the novel observations on the cell type-specific metabolic changes in diabetes that contribute to retinopathy. However reviewers expressed both unique and overlapping concerns that should be resolved in a revised manuscript. Namely, Reviewer 1 sought greater clarity on the timeline of observed transcriptional changes, as well as validation that levels of the glutamine transporter Slc38a3 are reduced in STZ retinas. This reviewer further offered multiple suggestions to improve both text and figure clarity. Reviewer 2 requested clarification on statistics and gene sets that inform pathway analysis, and further requested analysis based on specific cell populations. Here we concur in particular for data in Figure 3, where it is unclear if pathways presented relate to the bulk cell population or specific subsets. Reviewer 3 was overall enthusiastic but requested clarification on glucose metabolism by Muller cells, echoing a concern from Reviewer 1 (point 5), and we concur this should be addressed experimentally in a manner of your choosing. We invite you to submit a revised manuscript addressing the Reviewer comments noted here. While a revision must include responses to all reviewer points in some manner, additional data beyond those mentioned above are not required in a revised manuscript.

Thank you for this interesting contribution to Life Science Alliance. We are looking forward to receiving your revised manuscript.

Sincerely,

- A letter addressing the reviewers' comments point by point.
- An editable version of the final text (.DOC or .DOCX) is needed for copyediting (no PDFs).
- High-resolution figure, supplementary figure and video files uploaded as individual files: See our detailed guidelines for

preparing your production-ready images, <https://www.life-science-alliance.org/authors>

B. MANUSCRIPT ORGANIZATION AND FORMATTING:

Reviewer #1 (Comments to the Authors (Required)):

The authors investigate the transcriptional changes that occur in the retina and retinal endothelial cells during diabetic retinopathy progression using a STZ-induced diabetes model. They performed bulk RNA sequencing on retina and retinal endothelial cells at 1, 3, and 6 months, and single cell RNA sequencing on retinal tissues from control and diabetic mice. These data are supported by IHC staining of mouse and human retinal tissue, RT-qPCR validation of gene dysregulation, and analysis of gene expression and metabolic changes using ex vivo retinal tissue and cultured cell lines. The study highlights transcriptional changes affecting processes that may contribute to DR pathology, such as inflammation, Muller cell glutamine metabolism, and fibrosis.

The data presented in this study are useful, and contribute to the understanding of DR pathology. However, the manuscript may be improved substantially by addressing the following concerns:

General concerns:

1. A core theme of the article is establishing a timeline of transcriptional changes that occur in the retina and retinal endothelial cells during DR progression. However, the sequence of transcriptional changes isn't well presented in the figures and is not discussed explicitly in the article. Authors should consider more directly comparing the selection of DEGs at each timepoint - are there any that are consistently dysregulated? How do the uniquely dysregulated genes at each timepoint relate to disease progression as it is currently understood?
2. SLC38A3 change is important for the proposed model, but the result in Fig 3 is not convincing. An IHC to show the distribution of SLC38A3 in STZ retinas will strengthen the model.
3. Some of the figures are confusingly laid out and unclearly formatted (elaborated below). Consider using a consistent colouring scheme across figures.
4. Several statements are made throughout the manuscript which should be supported by the inclusion of a reference.
5. It is asserted that Muller cells upregulate glutaminolysis in high glucose conditions for anaplerosis to fulfil bioenergetic demands. However, it is not clear why glucose is not instead used as a substrate for TCA cycle and OXPHOS. Glutamine may instead be used as a source for biosynthesis of other amino acids and proteins for example synaptic proteins.
6. Some of the nomenclature is inconsistent in figures and the text, for example, the use of the symbol ' α ' vs 'alpha', aminoacid vs amino acid.
7. The discussion section is very long and would benefit from editing down to perhaps 50% of its current length.
8. The article, particularly in the results and discussion sections, could benefit from additional proofreading to improve grammar. There are several parts of these sections that are awkwardly phrased and difficult to read.

Specific concerns:

- Title: Should be 'links'
- Page 2: Define DRN
- Page 3: "To study the consequences of diabetes on the entire retina we performed transcriptomics analyses during the first 6 months of diabetes in the STZ mouse model to understand the pathways that drive the vascular and neural degeneration in diabetes." How does this tell you these pathways are 'drivers'?
- Page 4: "qPCR of endothelial-enriched lysates confirmed upregulation of H2Aa, H2Ab1, Gbp6, and a trend for Ilgp1 (Figure 2C)." What is the function of these genes and why were they singled out for confirmation by qPCR?

- Page 5: "Upregulated pathways included G alpha and cAMP signaling (Figure 2H), as well as GPCRs that mediate downstream effects such as VEGF activation." This is phrased a bit oddly given that G alpha and GPCR signalling are very closely associated. It's uncertain what the relevance of VEGF is here? If the point is the association with DR, then this could be explained more explicitly, perhaps with a reference. VEGF is not mentioned elsewhere in the manuscript. It might be more relevant to link GPCR signalling with Rho and RhoA given that rhodopsin is a GPCR.
- Page 5: "Cacnb2 encodes the $\beta 2$ subunit of the voltage-gated Ca^{2+} channel ($\text{Cav}\beta 2$) in photoreceptor ribbon synapses". It would be helpful to briefly explain what ribbon synapses are and relevance of calcium channels in this context (ribbon synapses are explained in more detail in the discussion, but could be briefly explained here to give the reader context).
- Page 5: "Human donor eyes with diabetic retinopathy (DR) exhibited similar disorganization and reduction in PSD95 and RIBEYE in the macular outer plexiform layer (OPL) when compared to healthy controls (Figure 3G)." This is confusing because you have already defined DR in the introduction, but the human DR images are labelled as 'DM' in Fig. 3G.
- Page 5/6: "Glutamine is physiologically produced by glial cells, especially Müller cells, and shuttled into the photoreceptors, where it is converted into glutamate, the main excitatory neurotransmitter in the photoreceptor ribbon synapses, and an important molecule for photoreceptor metabolism and survival." This information warrants a reference.
- Page 6: "... and confirming persistent photoreceptor decline from the 1 month timepoint." Is this really accurate given the number of DEGs at 3 months was a small fraction of the DEGs at 1 month according to the bulk RNA seq?
- Page 6: "Glul gene codifies Glutamine Synthase (GS), the enzyme responsible for glutamine production." ...from glutamate
- Page 7: "We then performed immunofluorescence in retinal cross-sections and confirmed significant decrease in GS expression in the inner nuclear layer (at the Müller cell level) in STZ when compared to sham retinas, as shown in the Figure S5." IHC staining for GLAST would be helpful here.
- Page 7: "Based on these data we hypothesize that, in the setting of high glucose, Müller cells switch to anaplerosis, utilizing intracellular glutamine for their own energetic needs, and, instead of removing the glutamate..." Why would glutamine be preferred over glucose for energetic needs?
- Page 8: "...thereby depriving them of this essential amino acid." Avoid using 'essential' amino acid here because essential amino acids are a defined class. Also, be consistent about amino acid vs amino acid.
- Page 8: "...trend of increased glutamate levels in the retinal tissue...". This is debatable.
- Page 8: "Under diabetic condition, Müller cell dysfunction and reduction of GLAST expression impairs this uptake, leading to an accumulation of glutamate in retinal neurons." Why would impaired uptake of glutamate by Müller cells lead to accumulation of glutamate inside retinal neurons? Wouldn't it accumulate extracellularly? Moreover, how do you reconcile decreased GLAST expression with decreased glutamate from the media?
- Page 9: "...and observed downregulation of mitochondrial genes (mt-Nd1, mt-Nd2, mt-Nd3, mt-Co3)...". Should mention these are ETC genes.
- Page 9: "Since endothelial cells rely on glutamine, provided by Müller cells, for their metabolism..." This statement warrants a reference.
- Page 9: "We did not observe any change in the glutamine uptake "receptors", Slc38a2, Slc38a3..." *Transporters?
- Page 9: "We wondered if scRNA-Seq differentially under-sampled "diseased" endothelial cells, obscuring their gene expression changes." It is not clear how this occurs, and how analyzing the bulk RNA seq data addresses the issue. May be worth explaining this further.
- Page 9: "In the absence of nutrients and amino acids, such as glutamine, mTOR is deregulated together with other metabolic pathways" Yes but mTOR is typically regulated at the protein level by posttranslational modification and not the level of mTOR gene transcription. While it is possible that mTOR gene transcription may be influenced by nutrient conditions, this mechanism is not canonical and should be discussed in further detail.
- Page 10: "Since mTOR reduction is associated with starvation and subsequent increase in autophagy, we..." What is meant by 'mTOR reduction?', decreased mTOR activity or decreased mTOR gene transcription? Either way, this should have a reference.
- Page 10: "...we next analyzed autophagic gene expression and found no significant alteration of ATG5 and BCLN1." This is unsurprising given that mTOR regulates autophagy through its kinase activity (e.g. ULK1 phosphorylation), not transcriptional regulation of autophagy genes.
- Page 10: "...the TCA cycle gene expression in endothelial cells, and..." What TCA cycle genes? Only ETC and glutaminolysis genes are downregulated.
- Page 11: "...we examined the timeline of retinal molecular changes in a mouse model of diabetes over the course of 6 months." *Retinal transcriptional/gene expression changes.
- Page 11: "Our results highlight the 3-month timepoint and Müller cells as particularly critical in the pathophysiology of DRD". This claim might need to be reconsidered. The authors state on page 5 that the 3-month timepoint was only investigated further with scRNA seq because there was a lack of DEGs in the bulk RNA seq. Moreover, it isn't clear that Müller cells possess any more DEGs in DR than any other retinal cell type. While the transcriptional and metabolic changes that Müller cells undergo in high glucose and DR are interesting, the way the data is presented implies that the Müller cells were selected for closer analysis more or less arbitrarily.
- Page 12: "Müller cells are normally deficient in glycolytic enzymes, and instead rely on lactate released from the photoreceptors (and endothelial cells) to power their TCA cycle (Figure 7)." This should have a reference.
- Page 12: "The photoreceptors are the closest cell to the choroidal glucose source..." Strictly speaking RPE cells are.
- Page 12: "...instead of being donated to other cells, is retained, which we confirmed by ex vivo retinal culture." This hypothesis was supported, rather than confirmed by, the ex vivo retinal culture data.
- Page 12: "...confirming diminished glutamine production by Müller cells." This is an overinterpretation.
- Page 12: "...upregulation of the glycolytic enzymes Pfkfb3 and Ldha in the Müller cells also suggested relative activation of

glycolysis, which is unusual for Müller cells." This is not shown in the figures. It is also at odds with the hypothesis that Müller cells are breaking down glutamine to fuel the TCA cycle.

- Fig 12: |The aKG produced by the Müller cell TCA cycle is then converted to glutamate (via the GDH enzyme) and then to glutamine (GS enzyme), which is made available to other cells for their TCA cycle and metabolic needs". This is not accurate. Glutamate is mostly produced through transaminase reaction rather than GDH.
- Page 13: "These findings were previously shown in..." It is unclear which findings are being referred to.
- Page 13: "as well as activate glutathione synthetase, the only enzyme capable of reducing retinal ammonia (Kobat and Turgut, 2020; Li and Puro, 2002)". This statement is not correct: 1) glutathione synthetase could not remove ammonia; 2) it is not the only enzyme to remove ammonia. There are many enzymes in the retina that can remove ammonia. For example, glutamine synthetase, asparagine synthetase, glutamate dehydrogenase and urea cycle enzymes (low expression in the retina).
- Page 15: "Our cultured HRMEC experiments..." Define HRMEC
- Page 18: "In addition, we acknowledge the endothelial cell dataset had photoreceptor contamination on bulkRNA-Seq, so we focused on the endothelial cell-specific gene signatures and excluded any photoreceptor-specific genes." This was not mentioned previously. This seems like an odd place to mention it for the first time.
- Page 19: "our study revealed Müller cell (macroglial) metabolic dysregulation and glutamine starvation as central drivers of endothelial and neuronal degeneration,.." This is a bold claim that is not adequately supported by the data presented.
- Page 19/20: when were blood glucose measurements taken with regards to feeding cycle?
- Page 20: "For dissociation, dissected retinas were transferred to a 500µL solution of 0.22µm filter sterilized 1x Dulbecco's Phosphate Buffered Saline (DPBS).." should be µm
- Page 20: "Samples were then spun down at 400g for 8 minutes at 40C and cell pellets were..." should be 400 x g
- Page 25: "MIO-M1 cells were cultured in DMEM supplemented with 10% FBS and antibiotics (1% P/S). Cells were cultured in 30 mM D Glucose or 24.6 mM Mannitol for 24 hours and 48 hours." The 30 mM glucose was the high glucose treatment? This also implies that the low glucose treatment was normal DMEM, which is usually ~25 mM glucose. Physiological levels are ~5 mM. Product catalog would be helpful. Also, why the difference between glucose and mannitol concentrations?
- Page 22 and 39: Define: BACT

Figures:

- Figure 2. Layout is very confusing. Upregulated pathways in the retina based on enrichment analysis seems to be present in both 2B and 2H. Would be helpful if endothelial and retina panels were grouped together, ordered consistently, and labelled with 'retina' or 'endothelial cells' or similar. Figure legend is incorrect (I think 2B is supposed to be endothelial cells, for example). 2C and 2G, should give the housekeeping gene in the legend.
- Figure 3. Thresholds are not defined in panels 3B and 3D. Would be helpful if these panels were labelled as 'Rods' and 'Cones'. Consider highlighting Cacnb2. Define CTF in 3F. Define BACT in 3I.
- Figure 4. High and low glucose concentrations should be defined for 4E and other panels either in the legend or on the figure. In 4E, the y-axis label for glutamine quantification in cell lysates is 'nM glutamine/cell number' which doesn't make sense. Also, the methods section suggests that intracellular glutamate also normalized to cell count but this is not reflected in the figure. 4F: should give housekeeping gene in the legend.
- Figure 6. In panels 6I and 6H the controls should be presented first to be consistent with previous figures.
- Figure 7. GLS is expressed more in the neurons than the glia. The hypothesis of upregulated Glutaminolysis inside Müller glia is not convincing from the data. Furthermore, GLUD1 activation has very high km to activate comparing to transaminase. The GLUD1 will produce or release ammonia which is not reflected in the figure. SLC38A3 downregulation in photoreceptors and endothelial cells need validation.
- Figure 7. A core feature of the article is establishing a 'timeline' of transcriptional events during DR progression. Consider reflecting this in the figure as the retinal/endothelial transcriptomes vary quite a bit at each timepoint.

Reviewer #2 (Comments to the Authors (Required)):

Scheri et al. investigated early molecular changes in diabetic retinas using both bulk and single-cell RNA sequencing. Their findings were validated across multiple models, including ex vivo retina tissue, cultured human Müller glia and endothelial cells, and retinal cross-sections from diabetic patients. The studies are important and will fill our current knowledge gap of early diabetic retinopathy.

However, there is a major concern regarding the pathway analysis presented in the manuscript. The authors relied on raw p-values to identify enriched pathways. In pathway enrichment analysis, multiple comparisons are made simultaneously, increasing the likelihood of false positives. Adjusted p-values, such as those calculated using the Benjamini-Hochberg method, are essential to control the false discovery rate and ensure the reliability of reported pathways.

A preliminary review of the gene list provided in "4261_0_data_set_3410634_sxpptn" did not replicate many of the pathways highlighted in the manuscript. The authors are encouraged to revisit the analysis using adjusted p-values and consider refining the interpretation and hypothesis accordingly.

For the single-cell RNA-seq data, only a subset of genes was presented. To provide a more comprehensive understanding of cell-type-specific changes, pathway analysis is recommended for each individual cell cluster. Additionally, the original dataset should be deposited in a public repository, and the corresponding accession code should be included in the manuscript.

Reviewer #3 (Comments to the Authors (Required)):

In this manuscript Scheri et. al. looked at metabolic relation between muller cells, photoreceptor and endothelial cells in DR. They used cell culture model, explants, STZ induced diabetes mouse model, and human tissue to measure these changes. Their work demonstrates that the 3 months' time point is the point of inflection where metabolism is re-wired to faulty anaplerosis leading to photoreceptor and endothelial dysfunction. This is a very impressive piece of work; however, authors must address these minor issues:

N=2 for Sham in figure 4C is not acceptable. Please have at least an n of 3. Assumption of normality can be made with at least n=3, and DF=2.

Authors claim, "Although Müller cells are the main GS expressing cells in the retina, GS as well as GLAST are expressed by other cells such as astrocytes (Figure 4D)."

Figure 4D shows 4 clusters of Muller cells. I don't see any Astrocyte data in Figure 4D.

Figure S5 IHCs labeled with GS are assumed to have less GS expression. Since GS is a Muller cell marker this can also simply mean less Muller cells in STZ mice. Can authors co-label these sections with CRALB or another marker (may be RLBP1) and then demonstrate GS is low in STZ treated whereas Muller cell number is not affected?

In the cell culture method section, authors state "MIO-M1 cells were cultured in DMEM supplemented with 10% FBS and antibiotics (1% P/S). Cells were cultured in 30 mM D-Glucose or 24.6 mM Mannitol for 24 hours and 48 hours"

Authors may have forgotten to write glutamine in their MIO-M1 method section, or this may be produced from glucose in their culture conditions. Please clarify. If there was no lactate or pyruvate or aspartate in their DMEM and no glutamine was supplemented, why do they see close to 45nM glutamine in spent media (Figure 4E)? Authors have used lactate and aspartate in their explants culture which they reasoned to be necessary for glutamine production by Muller cells.

Authors have selectively discussed scRNA seq data in their results, for example mt-Nd1, mt-Nd2, and mt-Nd3 changed in both muller cells (figure4) and retinal endothelial cells (figure5), however, these changes were only discussed for endothelial cells in the results section not muller cells. Can authors speculate on why do they see these changes in both cell types? Can these changes precede anaplerotic switch in Muller cells?

Typo on page 3- EEAT1 must be replaced with EAAT1.

Authors have stated multiple times in the manuscript that the muller cells don't have glycolytic enzymes. There is good enough expression of most of the rate limiting glycolytic enzymes in the muller cells in human protein atlas single cell data set.

Additionally, authors have used lactate and aspartate free media in their MIO-M1 cultures and were able to measure glutamine production rate. Where is this glutamine coming from? Can this be glycolytic in nature?

Please check and plot the expression of hexokinase, phosphofructokinase and pyruvate kinase in muller cells from your scRNAseq data and compare with photoreceptors and/or endothelial cells to prove that Muller cells lack these glycolytic enzymes.

We thank the editor and the reviewers for the helpful suggestions and comments. We thoroughly addressed the questions raised and hereby provide responses to the comments.

Reviewer #1:

The authors investigate the transcriptional changes that occur in the retina and retinal endothelial cells during diabetic retinopathy progression using a STZ-induced diabetes model. They performed bulk RNA sequencing on retina and retinal endothelial cells at 1, 3, and 6 months, and single cell RNA sequencing on retinal tissues from control and diabetic mice. These data are supported by IHC staining of mouse and human retinal tissue, RT-qPCR validation of gene dysregulation, and analysis of gene expression and metabolic changes using ex vivo retinal tissue and cultured cell lines. The study highlights transcriptional changes affecting processes that may contribute to DR pathology, such as inflammation, Muller cell glutamine metabolism, and fibrosis. The data presented in this study are useful and contribute to the understanding of DR pathology. However, the manuscript may be improved substantially by addressing the following concerns:

General concerns:

1. A core theme of the article is establishing a timeline of transcriptional changes that occur in the retina and retinal endothelial cells during DR progression. However, the sequence of transcriptional changes isn't well presented in the figures and is not discussed explicitly in the article. Authors should consider more directly comparing the selection of DEGs at each timepoint - are there any that are consistently dysregulated? How do the uniquely dysregulated genes at each timepoint relate to disease progression as it is currently understood?

R. Our supplemental data (Figure S2) demonstrate a small set of genes (and pathways) that are consistently up- or downregulated across the different time points, while many other genes show time point-specific changes. The uniquely dysregulated genes identified at each time point likely reflect distinct molecular and cellular processes that correspond to different stages of diabetic retinopathy progression. Early time points highlight genes related to the stress response, inflammation, or vascular dysfunction, while later time points reveal genes associated with neurodegeneration, metabolic alterations, tissue remodeling. This pattern aligns well with the known progression of diabetic retinopathy (Feit-Leichman et al. 2005; Sadikan et al. 2023) where early stages are characterized by inflammation, followed by neurodegeneration, and later by blood vessel loss and fibrosis.

2. SLC38A3 change is important for the proposed model, but the result in Fig 3 is not

convincing. An IHC to show the distribution of SLC38A3 in STZ retinas will strengthen the model.

R. We thank the reviewer for this insightful comment. We agree that demonstrating the spatial distribution of SLC38A3 would add important support to our model. To address concerns regarding the robustness of the result shown in Figure 3 (now split in figure 3 and 4), we have repeated the Western blot analysis using additional independent STZ and control retinal samples. The updated data, now included in the revised Figure 4, confirm the observed changes in SLC38A3 expression and reinforce our initial findings. In parallel, we performed immunohistochemistry (IHC) to assess the localization of SLC38A3 in STZ retinas and found a significant reduction in the IS (inner segment) of photoreceptors in the STZ retinas when compared to Sham. The data are included in the manuscript.

3. Some of the figures are confusingly laid out and unclearly formatted (elaborated below). Consider using a consistent coloring scheme across figures.

R. We appreciate the reviewer's feedback regarding figure clarity and formatting. In response, we have carefully revised the layout of the figures to improve readability and visual consistency. Specifically, we made sure we have adopted a consistent color scheme across all figures to aid interpretation and comparison, figure panels have been rearranged where necessary to ensure logical flow and alignment with the text; we checked that font sizes, labels, and legends are consistent for clarity. We have also updated figure captions to provide clearer guidance for the reader.

4. Several statements are made throughout the manuscript which should be supported by the inclusion of a reference.

R. We thank the reviewer for this important observation. We have carefully reviewed the manuscript and identified statements that required additional references for support.

5. It is asserted that Muller cells upregulate glutaminolysis in high glucose conditions for anaplerosis to fulfil bioenergetic demands. However, it is not clear why glucose is not instead used as a substrate for TCA cycle and OXPHOS. Glutamine may instead be used as a source for biosynthesis of other amino acids and proteins for example synaptic proteins.

R. We appreciate the reviewer's insightful comment regarding the metabolic role of glutaminolysis in Müller cells under high glucose conditions. While glucose is indeed a primary substrate for the TCA cycle and oxidative phosphorylation, Müller cells exhibit low expression of key glycolytic enzymes such as pyruvate kinase (PK) compared to

other retinal cells, limiting their capacity to efficiently metabolize glucose through glycolysis and feed the TCA cycle (Lindsay, Hurley et al 2014). This metabolic characteristic, combined with additional metabolic stress imposed by high glucose environments, such as oxidative stress and mitochondrial dysfunction, further impairs glucose utilization for energy production. Under physiological conditions, Müller cells preferentially rely on alternative substrates, including lactate and aspartate supplied by photoreceptors and retinal neurons, to fuel the TCA cycle (Lindsay, Hurley et al 2014). However, in diabetic or high glucose stress states, the availability or utilization of these substrates may be compromised, or energetic and biosynthetic demands may exceed what these fuels alone can support. In our bulk RNA-Seq we have found alteration of lactate transport at 3 months of diabetes, consistent with metabolic impairment in the diabetic retina. Müller cells may increase glutaminolysis to support anaplerosis and maintain TCA cycle intermediates, especially if glucose metabolism is impaired. Additionally, as the reviewer suggests, glutamine could serve also as a critical substrate for biosynthesis, including amino acids and proteins such as synaptic components, which may be essential for cellular adaptation and survival under stress.

6. Some of the nomenclature is inconsistent in figures and the text, for example, the use of the symbol ' α ' vs 'alpha', aminoacid vs amino acid.

R. We thank the reviewer for pointing out the inconsistencies in nomenclature. We have now thoroughly reviewed the manuscript and all figure labels to ensure consistency in terminology and formatting.

7. The discussion section is very long and would benefit from editing down to perhaps 50% of its current length.

R. We have carefully revised and streamlined this section to improve clarity and focus. Specifically, we have removed repetition and background information already covered in the Introduction, condensed or rephrased extended interpretations to be more concise, focused the discussion more tightly on key findings and their implications.

8. The article, particularly in the results and discussion sections, could benefit from additional proofreading to improve grammar. There are several parts of these sections that are awkwardly phrased and difficult to read.

R. We thank the reviewer for highlighting this issue. In response, we have thoroughly revised the manuscript to improve grammar, clarity, and overall readability. We have carefully edited awkward or ambiguous phrasing and ensured that the narrative flows more smoothly. We hope these revisions have significantly enhanced the clarity and

presentation of our work.

Specific concerns:

- Title: Should be 'links'

R. We thank the reviewer for pointing this out. The title has been corrected.

- Page 2: Define DRN

R. The abbreviation DRN has now been defined at its first occurrence in the manuscript. We have also ensured consistent use of this abbreviation throughout the text.

- Page 3: "To study the consequences of diabetes on the entire retina we performed transcriptomics analyses during the first 6 months of diabetes in the STZ mouse model to understand the pathways that drive the vascular and neural degeneration in diabetes." How does this tell you these pathways are 'drivers'?

R. We thank the reviewer for this important observation regarding our interpretation. We agree that transcriptomics analyses alone do not definitively establish causality or identify pathways as direct “drivers” of vascular and neural degeneration. Our intention was to highlight that the identified pathways are significantly associated with disease progression and likely contribute to these degenerative processes. To clarify, we have revised the sentence to avoid implying causality and now describe these pathways as “potential contributors” or “associated pathways” involved in vascular and neural degeneration. We believe this wording more accurately reflects the data and avoids overstating our conclusions.

- Page 4: "qPCR of endothelial-enriched lysates confirmed upregulation of H2Aa, H2Ab1, Gbp6, and a trend for Ilgp1 (Figure 2C)." What is the function of these genes and why were they singled out for confirmation by qPCR?

R. These genes were selected for qPCR validation because they were among the most significantly upregulated immune-related genes in our endothelial transcriptomic data. H2-Aa and H2-Ab1 are key components of the MHC class II complex, involved in antigen presentation and indicative of immune activation. Gbp6 is an interferon-inducible gene associated with innate immune responses, and Ilgp1 (also interferon-inducible) plays a role in immune signaling. We have added context in the manuscript to explain the biological relevance of these genes and the rationale for their selection.

- Page 5: "Upregulated pathways included G alpha and cAMP signaling (Figure 2H), as

well as GPCRs that mediate downstream effects such as VEGF activation." This is phrased a bit oddly given that G alpha and GPCR signalling are very closely associated. It's uncertain what the relevance of VEGF is here? If the point is the association with DR, then this could be explained more explicitly, perhaps with a reference. VEGF is not mentioned elsewhere in the manuscript. It might be more relevant to link GPCR signalling with Rho and Rdh8 given that rhodopsin is a GPCR.

R. We thank the reviewer for this helpful observation. We agree that the original phrasing was unclear, particularly in separating G alpha and GPCR signaling, which are functionally linked. We have revised the text to reflect this relationship more accurately. The inclusion of VEGF in this context is meant to highlight how GPCR signaling pathways can lead to VEGF activation and thus contribute to DR pathology. To strengthen this point, we have added a citation and expanded the discussion in the relevant section to make this link more explicit.

- Page 5. "Cacnb2 encodes the $\beta 2$ subunit of the voltage-gated Ca^{2+} channel ($\text{Cav}\beta 2$) in photoreceptor ribbon synapses". It would be helpful to briefly explain what ribbon synapses are and relevance of calcium channels in this context (ribbon synapses are explained in more detail in the discussion but could be briefly explained here to give the reader context).

R. To improve reader understanding, we have added a brief explanation of ribbon synapses and the role of calcium channels in photoreceptor neurotransmission in the Results section, where *Cacnb2* is first mentioned. This provides essential context prior to the more detailed discussion later in the manuscript.

- Page 5. "Human donor eyes with diabetic retinopathy (DR) exhibited similar disorganization and reduction in PSD95 and RIBEYE in the macular outer plexiform layer (OPL) when compared to healthy controls (Figure 3G)." This is confusing because you have already defined DR in the introduction, but the human DR images are labelled as 'DM' in Fig. 3G.

R. We thank the reviewer for pointing out this inconsistency. To avoid confusion, we have now standardized the labeling throughout the manuscript and figures, using "DR" consistently to refer to diabetic retinopathy, including in Figure 3G. This correction is reflected in the revised manuscript and all figure labels.

- Page 5/6: "Glutamine is physiologically produced by glial cells, especially Müller cells, and shuttled into the photoreceptors, where it is converted into glutamate, the main excitatory neurotransmitter in the photoreceptor ribbon synapses, and an important

molecule for photoreceptor metabolism and survival." This information warrants a reference.

R. We thank the reviewer for highlighting this. We have now added relevant references to support the statement regarding glutamine production by Müller cells and its role in photoreceptor glutamate metabolism. These citations have been included at the appropriate point in the text.

• Page 6: "... and confirming persistent photoreceptor decline from the 1-month timepoint." Is this really accurate given the number of DEGs at 3 months was a small fraction of the DEGs at 1 month according to the bulk RNA seq?

R. We thank the reviewer for pointing this out. While it is correct that the number of DEGs at 3 months was relatively limited in the bulk RNA-seq data, our single-cell RNA-seq analysis at the same time point revealed differentially expressed genes specifically in photoreceptors. This suggests that photoreceptor dysregulation persists at 3 months but may not be readily detected in bulk RNA-seq at that timepoint.

• Page 6: "Glul gene codifies Glutamine Synthase (GS), the enzyme responsible for glutamine production." ...from glutamate

R. We thank the reviewer for this clarification. We have revised the sentence to accurately reflect that Glutamine Synthase (GS) catalyzes the production of glutamine from glutamate.

• Page 7: "We then performed immunofluorescence in retinal cross-sections and confirmed significant decrease in GS expression in the inner nuclear layer (at the Müller cell level) in STZ when compared to sham retinas, as shown in the Figure S5." IHC staining for GLAST would be helpful here.

R. We agree that GLAST immunohistochemistry would provide important complementary information regarding Müller cell function. We performed GLAST staining on retinal cross-sections, as suggested, and we observed a significant decrease of GLAST at the INL level of STZ retinas, where the Muller cell bodies are. We included these results in the manuscript (Figure S5).

• Page 7: "Based on these data we hypothesize that, in the setting of high glucose, Müller cells switch to anaplerosis, utilizing intracellular glutamine for their own energetic needs, and, instead of removing the glutamate..." Why would glutamine be preferred over glucose for energetic needs?

R. We thank the reviewer for this important concern. While glucose is indeed a primary substrate for the TCA cycle and oxidative phosphorylation, our data suggest that under diabetic stress, Müller cells may increase glutaminolysis to support anaplerosis and maintain TCA cycle intermediates. As mentioned above, Müller cells have low expression of key glycolytic enzymes such as pyruvate kinase (PK) compared to other retinal cells, limiting their capacity to efficiently metabolize glucose through glycolysis and feed the TCA cycle (Lindsay, Hurley et al 2014). Under physiological conditions, Müller cells preferentially rely on alternative substrates, including lactate and aspartate supplied by photoreceptors and retinal neurons, to fuel the TCA cycle (Lindsay, Hurley et al 2014).

- Page 8: "...thereby depriving them of this essential aminoacid." Avoid using 'essential' amino acid here because essential amino acids are a defined class. Also, be consistent about aminoacid vs amino acid.

R. We thank the reviewer for this important clarification. We have revised the manuscript to avoid using "essential" in this context and corrected the term to "amino acid." Additionally, we conducted a thorough review to ensure consistent use of "amino acid" throughout the manuscript and figures.

- Page 8: "...trend of increased glutamate levels in the retinal tissue...". This is debatable.

R. We appreciate the reviewer's comment. Based on our measurements, there is a trend toward increased glutamate levels in the retinal tissue under diabetic conditions, although it does not reach statistical significance (Figure 5). We have clarified this in the text by specifying that it is a non-significant trend. We agree that the interpretation should be cautious, and we have adjusted the wording accordingly to reflect this more accurately.

- Page 8: "Under diabetic condition, Müller cell dysfunction and reduction of GLAST expression impairs this uptake, leading to an accumulation of glutamate in retinal neurons." Why would impair uptake of glutamate by Muller cells lead to accumulation of glutamate inside retinal neurons? Wouldn't it accumulate extracellularly? Moreover, how do you reconcile decreased GLAST expression with decreased glutamate from the media?

R. We thank the reviewer for this insightful comment. The original phrasing was imprecise. We agree that impaired glutamate uptake by Müller cells due to reduced GLAST expression would primarily lead to extracellular accumulation of glutamate,

rather than accumulation inside retinal neurons. We have corrected this in the revised text to reflect that the accumulation occurs in the extracellular space, particularly around neurons, where it can contribute to excitotoxic stress.

Regarding the apparent discrepancy between decreased GLAST expression in Müller cells and reduced glutamate levels in the media, we speculate that other retinal cells are taking up the excess glutamate from the environment, especially under conditions where GLAST expression is reduced in Müller cells. To note that GLAST reduction was observed in Müller cell snRNA-Seq, cell culture and immunofluorescence in the INL where their cell bodies are suggesting a specific downregulation in the Müller cells only; the glutamate upregulation is observed in the ex vivo retinal culture where the whole retinal in cultured and not isolated Müller cells, and the latter is consistent with WB data from the whole retinal lysates.

- Page 9: "...and observed downregulation of mitochondrial genes (mt-Nd1, mt-Nd2, mt-Nd3, mt-Co3)...". Should mention these are ETC genes.

R. We thank the reviewer for this helpful suggestion. We have revised the manuscript to specify that these mitochondrial genes are components of the electron transport chain (ETC), thereby providing clearer context for their functional relevance.

- Page 9: "Since endothelial cells rely on glutamine, provided by Müller cells, for their metabolism..." This statement warrants a reference.

R. We thank the reviewer for highlighting this point. We have now added appropriate references to support the statement regarding the reliance of endothelial cells on glutamine supplied by Müller cells for their metabolism.

- Page 9: "We did not observe any change in the glutamine uptake "receptors", Slc38a2, Slc38a3..." *Transporters?

R. We thank the reviewer for this correction. We have revised the manuscript to replace "receptors" with "transporters" when referring to Slc38a2 and Slc38a3 to accurately reflect their function.

- Page 9: "We wondered if scRNA-Seq differentially under-sampled "diseased" endothelial cells, obscuring their gene expression changes." It is not clear how this occurs, and how analyzing the bulk RNA seq data addresses the issue. May be worth explaining this further.

R. Single-cell RNA-seq can under-represent certain cell populations, particularly those that are stressed, damaged, or undergoing apoptosis, as these cells may be more

fragile and less likely to survive the dissociation and capture process. This could result in an under-sampling of "diseased" endothelial cells, potentially obscuring transcriptomic changes associated with pathology. To address this, since we had bulk RNA-seq of endothelial-enriched lysates, we used this complementary approach to detect gene expression changes that may have been missed or attenuated in the single-cell dataset. We hope this explanation clarifies our rationale and addresses the reviewer's concern.

- Page 9: "In the absence of nutrients and amino acids, such as glutamine, mTOR is deregulated together with other metabolic pathways" Yes but mTOR is typically regulated at the protein level by posttranslational modification and not the level of mTOR gene transcription. While it is possible that mTOR gene transcription may be influenced by nutrient conditions, this mechanism is not canonical and should be discussed in further detail.

R. We thank the reviewer for this valuable observation. We agree that mTOR activity is primarily regulated through posttranslational modifications, particularly phosphorylation, in response to nutrient availability, rather than through changes in its transcription. Based on this insight and upon further analysis of our data, we recognized that mTOR is not significantly involved in the pathway we are describing. Accordingly, we have revised the relevant sections of the Results and Discussion.

- Page 10: "Since mTOR reduction is associated with starvation and subsequent increase in autophagy, we..." What is meant by 'mTOR reduction?', decreased mTOR activity or decreased mTOR gene transcription? Either way, this should have a reference.

R. In the manuscript, we were referring specifically to mTOR gene expression. However, we fully agree that mTOR activity is primarily regulated at the protein level via posttranslational modifications, particularly phosphorylation, in response to nutrient availability. To clarify this point, we have revised the text and added a brief discussion acknowledging that although mTOR transcription may be influenced under certain stress or nutrient-deprived conditions, this is not the predominant or canonical mechanism of its regulation. These changes are now reflected in the revised manuscript. We also added a reference.

- Page 10: "...we next analyzed autophagic gene expression and found no significant alteration of ATG5 and BCLN1." This is unsurprising given that mTOR regulates autophagy through its kinase activity (e.g. ULK1 phosphorylation), not transcriptional regulation of autophagy genes.

R. We agree that mTOR primarily regulates autophagy at the posttranslational level through its kinase activity, such as phosphorylation of ULK1, rather than through direct transcriptional control of autophagy-related genes. Our original analysis aimed to explore whether upstream nutrient signaling might also affect transcriptional regulation of key autophagy genes under our experimental conditions.

- Page 10: "...the TCA cycle gene expression in endothelial cells, and..." What TCA cycle genes? Only ETC and glutaminolysis genes are downregulated.

R. We thank the reviewer for this important clarification. Upon reevaluation, we recognize that the downregulated genes primarily involve components of the electron transport chain and glutaminolysis, rather than the broader TCA cycle. We have revised the manuscript accordingly to accurately reflect these findings.

- Page 11: "...we examined the timeline of retinal molecular changes in a mouse model of diabetes over the course of 6 months." *Retinal transcriptional/gene expression changes.

R. We have revised the sentence to specify "retinal transcriptional/gene expression changes" to more accurately describe the nature of the molecular changes examined.

- Page 11: "Our results highlight the 3-month timepoint and Müller cells as particularly critical in the pathophysiology of DRD". This claim might need to be reconsidered. The authors state on page 5 that the 3-month timepoint was only investigated further with scRNA seq because there was a lack of DEGs in the bulk RNA seq. Moreover, it isn't clear that Muller cells possess any more DEGs in DR than any other retinal cell type. While the transcriptional and metabolic changes that Muller cells undergo in high glucose and DR are interesting, the way the data is presented implies that the Muller cells were selected for closer analysis more or less arbitrarily.

R. We thank the reviewer for this insightful comment and appreciate the opportunity to clarify our rationale for focusing on the 3-month timepoint and Müller cells. As is well known from the progression of retinopathy in the STZ model, retinal vascular degeneration is manifest at 6mo, with loss of capillary. Therefore, we hypothesized that these changes would be heralded by transcriptional/molecular changes at the earlier timepoint of 3-months. Therefore, when the bulk RNA-seq revealed fewer differentially expressed genes (DEGs) overall compared to later timepoints, we were motivated to perform the single-cell RNA-seq, which provided higher resolution and uncovered critical transcriptional changes within specific retinal cell populations at this stage. Concerning our focus on Müller cells, we agree that they are one among several retinal cell types affected in diabetic retinopathy. However, we chose to investigate Müller cells

more deeply due to their well-established central role in retinal homeostasis and because their transcriptional and metabolic responses in our data suggested meaningful alterations relevant to disease progression. In addition to Müller cells, we also report on photoreceptors and endothelial cells at the same 3-month timepoint.

- Page 12: "Müller cells are normally deficient in glycolytic enzymes and instead rely on lactate released from the photoreceptors (and endothelial cells) to power their TCA cycle (Figure 7)." This should have a reference.

R. We thank the reviewer for this important suggestion. We have added appropriate references to support the statement regarding Müller cells' metabolic reliance on lactate from photoreceptors and endothelial cells.

- Page 12: "The photoreceptors are the closest cell to the choroidal glucose source..." Strictly speaking RPE cells are.

R. We thank the reviewer for this helpful correction. We acknowledge that, strictly speaking, the retinal pigment epithelium (RPE) cells are indeed the closest to the choroidal glucose source, with photoreceptors lying immediately adjacent to the RPE. In the revised version of the manuscript, we have slightly shortened the discussion, and this sentence was removed as part of that streamlining.

- Page 12: "...instead of being donated to other cells, is retained, which we confirmed by ex vivo retinal culture." This hypothesis was supported, rather than confirmed by, the ex vivo retinal culture data.

R. We thank the reviewer for this important clarification. We have revised the manuscript to state that this hypothesis was supported, rather than confirmed, by the ex vivo retinal culture data, reflecting the appropriate level of evidence.

- Page 12: "...confirming diminished glutamine production by Müller cells." This is an overinterpretation.

R. We thank the reviewer for this important observation. While our data do show a reduction in glutamine levels, we agree that attributing this solely to diminished production by Müller cells may overstate the evidence. To address this, we have revised the wording to more cautiously state that the findings are consistent with reduced glutamine availability, which may reflect altered production or metabolism involving Müller cells.

- Page 12: "...upregulation of the glycolytic enzymes *Pkm2* and *Ldha* in the Müller cells also suggested relative activation of glycolysis, which is unusual for Müller cells." This is not shown in the figures. It is also at odds with the hypothesis that Müller cells are breaking down glutamine to fuel the TCA cycle.

R. We acknowledge that the upregulation of *Pkm2* and *Ldha* in Müller cells is not directly depicted in the current figures and have updated the manuscript to reflect this (in the supplemental material). Furthermore, we acknowledge the apparent contradiction between glycolytic activation and the hypothesis of glutamine-driven TCA cycle metabolism in Müller cells. We propose that these metabolic pathways may coexist or represent a metabolic flexibility that Müller cells employ under diabetic conditions.

- Fig 12: |The aKG produced by the Müller cell TCA cycle is then converted to glutamate (via the GDH enzyme) and then to glutamine (GS enzyme), which is made available to other cells for their TCA cycle and metabolic needs". This is not accurate. Glutamate is mostly produced through transaminase reaction rather than GDH.

R. We agree that under normal physiological conditions, glutamate production primarily occurs via transaminase reactions. However, GDH activity may contribute to glutamate synthesis. We have now corrected the sentence highlighting this note.

- Page 13: "These findings were previously shown in..." It is unclear which findings are being referred to.

R. We thank the reviewer for pointing out this ambiguity. We have revised the sentence improving clarity for the reader.

- Page 13: "as well as activate glutathione synthetase, the only enzyme capable of reducing retinal ammonia (Kobat and Turgut, 2020; Li and Puro, 2002)". This statement is not correct: 1) glutathione synthetase could not remove ammonia; 2) it is not the only enzyme to remove ammonia. There are many enzymes in the retina that can remove ammonia. For example, glutamine synthetase, asparagine synthetase, glutamate dehydrogenase and urea cycle enzymes (low expression in the retina).

R. We thank the reviewer for this important correction. We agree that the statement regarding glutathione synthetase's role in ammonia removal was inaccurate. In the revised version of the manuscript, we have slightly shortened the discussion, and this sentence was removed as part of that streamlining.

- Page 15: "Our cultured HRMEC experiments..." Define HRMEC

R. We thank the reviewer for pointing this out. We have now defined HRMEC (human retinal microvascular endothelial cells) at the first mention in the manuscript to ensure clarity for all readers.

- Page 18: "In addition, we acknowledge the endothelial cell dataset had photoreceptor contamination on bulkRNA-Seq, so we focused on the endothelial cell-specific gene signatures and excluded any photoreceptor-specific genes." This was not mentioned previously. This seems like an odd place to mention it for the first time.

R. We thank the reviewer for this observation. We agree that this important methodological detail warrants earlier mention. We have now included a description of the photoreceptor contamination in the endothelial cell bulk RNA-Seq dataset and our approach to address it earlier in the Results to improve clarity and transparency.

- Page 19: "our study revealed Müller cell (macroglial) metabolic dysregulation and glutamine starvation as central drivers of endothelial and neuronal degeneration,..". This is a bold claim that is not adequately supported by the data presented.

R. We thank the reviewer for this important feedback. We agree that the wording of this statement may overstate the conclusions supported by our current data. We have revised the manuscript to soften the claim, emphasizing that our findings suggest a potential role for Müller cell metabolic dysregulation and glutamine deprivation in contributing to endothelial and neuronal changes, while acknowledging that further studies are needed.

- Page 19/20: when were blood glucose measurements taken with regards to feeding cycle?

R. Blood glucose measurements were taken at random time points, as the mice had ad libitum access to food. There was no defined feeding schedule, so measurements were not synchronized with specific feeding times. We added this in the method section of the manuscript.

- Page 20: "For dissociation, dissected retinas were transferred to a 500µL solution of 0.22um filter sterilized 1x Dulbecco's Phosphate Buffered Saline (DPBS)" should be µm

R. We thank the reviewer for pointing out this typo. We have corrected "0.22um" to "0.22 µm".

- Page 20: "Samples were then spun down at 400g for 8 minutes at 40C and cell pellets were..." should be 400 x g

R. We have updated the manuscript to specify centrifugation as "400 x g" for clarity.

- Page 25: "MIO-M1 cells were cultured in DMEM supplemented with 10% FBS and antibiotics (1% P/S). Cells were cultured in 30 mM D Glucose or 24.6 mM Mannitol for 24 hours and 48 hours." The 30 mM glucose was the high glucose treatment? This also implies that the low glucose treatment was normal DMEM, which is usually ~25 mM glucose. Physiological levels are ~5 mM. Product catalog would be helpful. Also, why the difference between glucose and mannitol concentrations?

R. We thank the reviewer for this important point regarding glucose concentrations. To clarify, we used a low glucose DMEM formulation (5.5 mM glucose, 1 g/L) to better represent physiological levels, and 30 mM glucose for the high glucose treatment. We have updated the Methods section to specify the exact DMEM formulations used, including catalog numbers, to improve transparency and reproducibility. The mannitol concentration (24.6 mM) corresponds to the difference between the high glucose (30 mM) and low glucose (5.5 mM) media, serving as an osmotic control to account for the increased osmolarity in the high glucose condition. This is now reported in the manuscript methods.

- Page 22 and 39: Define: BACT

R. We thank the reviewer for pointing this out. We have now defined the abbreviation "BACT" at its first occurrence in the manuscript for clarity.

Figures:

- Figure 2. Layout is very confusing. Upregulated pathways in the retina based on enrichment analysis seems to be present in both 2B and 2H. Would be helpful if endothelial and retina panels were grouped together, ordered consistently, and labelled with 'retina' or 'endothelial cells' or similar. Figure legend is incorrect (I think 2B is supposed to be endothelial cells, for example). 2C and 2G, should give the housekeeping gene in the legend.

R. We thank the reviewer for this detailed and helpful feedback regarding Figure 2. The figure panels are organized to group endothelial (top) and retinal (bottom) data separately and consistently. Clear labels have been added to indicate the cell type for each panel. The figure legend has been corrected to accurately describe each panel, including specifying that panel 2B corresponds to endothelial cells. Additionally, we have

added the housekeeping gene information to the legend for panels 2C and 2G as requested. These changes improve the clarity and interpretability of the figure.

- Figure 3. Thresholds are not defined in panels 3B and 3D. Would be helpful if these panels were labelled as 'Rods' and 'Cones'. Consider highlighting *Cacnb2*. Define CTF in 3F. Define BACT in 3I.

R. We thank the reviewer for these valuable suggestions. We have now defined the thresholds used in panels 3B and 3D in the figure legend. Panels 3B and 3D are now labelled as 'Rods' and 'Cones' respectively for clarity. Additionally, we have highlighted *Cacnb2* in the relevant panels as suggested. The abbreviations "CTF" in panel 3F and "BACT" in panel 3I have been defined in the figure legend. We hope these changes improve the clarity and interpretability of Figure 3.

- Figure 4. High and low glucose concentrations should be defined for 4E and other panels either in the legend or on the figure. In 4E, the y-axis label for glutamine quantification in cell lysates is 'nM glutamine/cell number' which doesn't make sense. Also, the methods section suggests that intracellular glutamate also normalized to cell count but this is not reflected in the figure. 4F: should give housekeeping gene in the legend.

R. We thank the reviewer for the valuable feedback. We have revised Figure 4 (now Figure 5) and its legend accordingly. The specific concentrations defining "high" and "low" glucose conditions are now clearly indicated in the figure legend to improve clarity.

Additionally, the intracellular glutamine and glutamate levels have been both normalized to cell count as described in the method section, and this is now clearly indicated in the figure legend and axis labels.

Finally, the housekeeping gene used for normalization in panel 4F (now 5F) has been added to the figure legend.

- Figure 6. In panels 6I and 6H the controls should be presented first to be consistent with previous figures.

R. We have updated panels 6H and 6I in Figure 6 (now Figure 8B), so that the control groups are now presented first, ensuring consistency with the layout of previous figures.

- Figure 7. GLS is expressed more in the neurons than the glia. The hypothesis of upregulated Glutaminolysis inside Muller glia is not convincing from the data. Furthermore, GLUD1 activation has very high K_m to activate comparing to transaminase. The GLUD1 will produce or release ammonia which is not reflected in the

figure. SLC38A3 downregulation in photoreceptors and endothelial cells need validation.

R. We appreciate the reviewer's insightful comments regarding GLS expression. In our experiments, we observed an increase in GLS specifically in Müller glia under high glucose conditions, which is reflected in now Figure 9 (previous Figure 7) (diabetic). Regarding GLUD1, we acknowledge the point about its higher Km and the potential for ammonia production (now reflected in the figure 9), which is not directly addressed in our current data. Additionally, we observed downregulation of SLC38A3 in photoreceptors by scRNA-Seq and in the whole retina samples by western blot, not endothelial cells, supporting the findings shown in Figure 9. We performed immunohistochemistry for SLC38A3 to further validate this observation and found a significant reduction in the IS (inner segment) of photoreceptors in the STZ retinas when compared to sham. The data are included in the manuscript.

- Figure 7. A core feature of the article is establishing a 'timeline' of transcriptional events during DR progression. Consider reflecting this in the figure as the retinal/endothelial transcriptomes vary quite a bit at each timepoint.

R. Thank you for this insightful suggestion. We agree that illustrating the timeline of transcriptional events during diabetic retinopathy progression would enhance the clarity of our findings. Figure 9 specifically reflects data from the 3-month time point, which is the crucial time point based on our data, and we have now clarified this in the figure legend and the main text.

Reviewer #2:

Scheri et al. investigated early molecular changes in diabetic retinas using both bulk and single-cell RNA sequencing. Their findings were validated across multiple models, including ex vivo retina tissue, cultured human Müller glia and endothelial cells, and retinal cross-sections from diabetic patients. The studies are important and will fill our current knowledge gap of early diabetic retinopathy.

However, there is a major concern regarding the pathway analysis presented in the manuscript. The authors relied on raw p-values to identify enriched pathways. In pathway enrichment analysis, multiple comparisons are made simultaneously, increasing the likelihood of false positives. Adjusted p-values, such as those calculated using the Benjamini-Hochberg method, are essential to control the false discovery rate and ensure the reliability of reported pathways.

A preliminary review of the gene list provided in "4261_0_data_set_3410634_sxpptn" did not replicate many of the pathways highlighted in the manuscript. The authors are

encouraged to revisit the analysis using adjusted p-values and consider refining the interpretation and hypothesis accordingly.

R. We thank the reviewer for this critical and well-founded comment. We acknowledge that relying solely on raw p-values in pathway enrichment analysis increases the risk of false positives due to multiple hypothesis testing. In response, we have reanalyzed the dataset using the Benjamini-Hochberg method to adjust p-values and control the false discovery rate. Based on this corrected analysis, we have updated the list of significantly enriched pathways. As a result of this more stringent approach, some pathways previously highlighted are no longer statistically significant and have been removed from the manuscript. We have revised the relevant interpretations and adjusted the discussion and conclusions accordingly to reflect these changes.

For the single-cell RNA-seq data, only a subset of genes was presented. To provide a more comprehensive understanding of cell-type-specific changes, pathway analysis is recommended for each individual cell cluster. Additionally, the original dataset should be deposited in a public repository, and the corresponding accession code should be included in the manuscript.

R. We thank the reviewer for this valuable suggestion. We did not perform pathway enrichment analysis on the single-cell RNA-seq data; our analysis of these data was limited to gene expression, which we presented using volcano (now showing significant changes in separated clusters) and violin plots. Pathway enrichment analyses were conducted solely on the bulk RNA-seq dataset.

Regarding data availability, we will deposit the raw data in GEO, and we will include the accession number upon publication, as stated in the manuscript section Data availability.

Reviewer #3:

In this manuscript Scheri et. al. looked at metabolic relation between muller cells, photoreceptor and endothelial cells in DR. They used cell culture model, explants, STZ induced diabetes mouse model, and human tissue to measure these changes. Their work demonstrates that the 3 months' time point is the point of inflection where metabolism is re-wired to faulty anaplerosis leading to photoreceptor and endothelial dysfunction. This is a very impressive piece of work; however, authors must address these minor issues:

N=2 for Sham in figure 4C is not acceptable. Please have at least an n of 3. Assumption of normality can be made with at least n=3, and DF=2.

R. We thank the reviewer for highlighting this important point regarding sample size. We have now added more samples to or analyses for both sham and STZ group, increasing

the sample size to n=5 for sham and 6 for STZ. The figures (now 4 and 5) and corresponding statistical analyses have been updated accordingly. This ensures appropriate assumptions regarding normality and strengthens the robustness of our findings.

Authors claim, "Although Müller cells are the main GS expressing cells in the retina, GS as well as GLAST are expressed by other cells such as astrocytes (Figure 4D)."

Figure 4D shows 4 clusters of Muller cells. I don't see any Astrocyte data in Figure 4D.

R. We agree that Figure 4D (now 5B) displays data from Müller cell clusters only and does not include astrocyte clusters. To clarify this, we have revised the manuscript text to remove the reference to astrocytes in relation to Figure 5B. Additionally, we now explicitly state that while GS and GLAST expression has been reported in astrocytes in the literature, the single-cell data presented here focus on Müller cell populations.

Figure S5 IHCs labeled with GS are assumed to have less GS expression. Since GS is a Muller cell marker this can also simply mean less Muller cells in STZ mice. Can authors co-label these sections with CRALB or another marker (may be RLBP1) and then demonstrate GS is low in STZ treated whereas Muller cell number is not affected?

R. As suggested by the reviewer, we also stained the retinal cross-section for CRALBP and we observed no changes in the expression and/or localization of CRALBP in diabetic retinas when compared to healthy ones, confirming selective decrease in GS compared other non-glutamine related Muller cell markers. The result is included in the manuscript (Figure S5).

In the cell culture method section, authors state "MIO-M1 cells were cultured in DMEM supplemented with 10% FBS and antibiotics (1% P/S). Cells were cultured in 30 mM D-Glucose or 24.6 mM Mannitol for 24 hours and 48 hours". Authors may have forgotten to write glutamine in their MIO-M1 method section, or this may be produced from glucose in their culture conditions. Please clarify. If there was no lactate or pyruvate or aspartate in their DMEM and no glutamine was supplemented, why do they see close to 45nM glutamine in spent media (Figure 4E)? Authors have used lactate and aspartate in their explants culture which they reasoned to be necessary for glutamine production by Muller cells.

R. We thank the reviewer for raising this important point. We would like to clarify that the MIO-M1 cell culture and retinal explant culture experiments were performed under distinct conditions.

MIO-M1 cells were cultured in standard DMEM, which includes both glutamine and pyruvate as part of its base formulation, along with high or normal glucose, 10% FBS, and antibiotics. The glutamine detected in the spent medium likely reflects a combination of: (1) residual glutamine present in the original medium, and (2) de novo synthesis via pyruvate- or glucose-driven TCA cycle activity generating α -ketoglutarate, a precursor for glutamate and subsequently glutamine via glutamine synthetase.

In contrast, retinal explants were cultured short-term in Krebs-Ringer HEPES-bicarbonate medium supplemented with glucose, lactate, and aspartate, following established protocols (Hurley et al. 2015). This was designed to mimic the in vivo metabolic environment, where Müller cells rely on photoreceptor-derived substrates to support glutamine synthesis and energy metabolism. While Müller cells retain the ability to metabolize glucose, this does not fully replicate the physiological metabolic coupling present in the intact retina. Therefore, glutamine synthesis observed in MIO-M1 cultures likely reflects a non-physiological, in vitro adaptation, whereas the explant model better represents the native substrate preferences of Müller cells in vivo.

Authors have selectively discussed scRNA seq data in their results, for example mt-Nd1, mt-Nd2, and mt-Nd3 changed in both muller cells (figure4) and retinal endothelial cells (figure5), however, these changes were only discussed for endothelial cells in the results section not muller cells. Can authors speculate on why do they see these changes in both cell types? Can these changes precede anaplerotic switch in Muller cells?

R. We agree that the mitochondrial gene changes observed in both Müller cells and retinal endothelial cells merit further discussion. These genes, encoding components of the mitochondrial respiratory chain (Complex I), likely reflect alterations in mitochondrial function and metabolic state. In endothelial cells, we highlighted these changes in the context of vascular metabolism, but similar mitochondrial gene expression alterations in Müller cells could indeed indicate a metabolic shift. It is plausible that these mitochondrial changes in Müller cells precede or accompany an anaplerotic switch, reflecting adaptation in their energy metabolism and substrate utilization to support retinal homeostasis under stress. We have now mentioned in the Results section to address this mitochondrial gene regulation in Müller cells, and speculate on the potential metabolic implications, including the possibility of an anaplerotic switch in Müller cells.

Typo on page 3- EEAT1 must be replaced with EAAT1.

R. We thank the reviewer for catching this typographical error. The text has been corrected to replace “EEAT1” with “EAAT1”.

Authors have stated multiple times in the manuscript that the muller cells don't have glycolytic enzymes. There is good enough expression of most of the rate limiting glycolytic enzymes in the muller cells in human protein atlas single cell data set. Additionally, authors have used lactate and aspartate free media in their MIO-M1 cultures and were able to measure glutamine production rate. Where is this glutamine coming from? Can this be glycolytic in nature?

R. We thank the reviewer for this thoughtful comment. While it is well-established that Müller cells primarily rely on lactate derived from photoreceptors and endothelial cells to fuel the TCA cycle under physiological conditions, we agree that recent single-cell RNA-seq datasets, including those from the Human Protein Atlas, show that Müller cells do express key glycolytic enzymes. This indicates that Müller cells retain glycolytic capacity and can metabolize glucose directly, particularly under stressed or artificial conditions such as in vitro culture.

In our MIO-M1 cell experiments, the culture medium (DMEM) contains both glutamine and pyruvate, in addition to high glucose, which can be metabolized via glycolysis. Therefore, the glutamine detected in the spent medium likely results from a combination of: (1) residual glutamine from the original medium, and (2) de novo synthesis via conversion of glucose or pyruvate to TCA cycle intermediates such as α -ketoglutarate, followed by glutamate and glutamine synthesis through glutamine synthetase.

While this demonstrates that MIO-M1 cells can produce glutamine in the absence of exogenous lactate and aspartate, this likely reflects an in vitro metabolic adaptation rather than the preferred in vivo pathway, where lactate and aspartate from photoreceptors play a more dominant role in Müller cell metabolism. The glutamine production observed in isolated Müller cells under glucose-supplemented conditions likely reflects a less physiologically relevant pathway compared to the lactate/aspartate-fueled glutamine synthesis observed in the ex vivo retina.

Please check and plot the expression of hexokinase, phosphofructokinase and pyruvate kinase in muller cells from your scRNAseq data and compare with photoreceptors and/or endothelial cells to prove that Muller cells lack these glycolytic enzymes.

R. We appreciate the reviewer's suggestion to examine the expression of key glycolytic enzymes such as hexokinase, phosphofructokinase, and pyruvate kinase in Müller cells compared to photoreceptors and endothelial cells using our single-cell RNA-seq data. We would like to clarify that we have not stated that Müller cells lack these glycolytic genes but that they predominantly rely on lactate supplied by neighboring photoreceptors and retinal endothelial cells as supported by literature (Linsday, Hurley,

2014). In our bulk RNA-Seq we have found alteration of lactate transport at 3 months of diabetes.

Under diabetic conditions, Müller cells may increase glutaminolysis to support anaplerosis and replenish TCA cycle intermediates, especially if glucose metabolism or symbiotic relationship with endothelial cells or photoreceptors are impaired.

In response to the reviewer's comment, we analyzed the expression of key glycolytic enzymes, including *Pkm*, *Ldha*, *Pfkm*, and *Hk1* in Müller cells, photoreceptors, and endothelial cells. We found that Müller cells express these enzymes, with a significant upregulation of *Pkm* and *Ldha* in a subset of Müller cells from STZ-treated retinas. This suggests an increased glycolytic activity in these cells under diabetic stress, likely reflecting their attempt to metabolize the glucose. Interestingly, we observed a significant increase of *Pkm* and *Ldha* in two clusters of cones and almost all the rods, suggesting an increase in glucose utilization in response to glutamine deprivation under hyperglycemia. We didn't observe any changes of these genes in retinal endothelial cells. We now include those plots in the manuscript as supplemental figures (S6).

November 3, 2025

RE: Life Science Alliance Manuscript #LSA-2025-03434-TR

Dr. Amani A. Fawzi
Northwestern University
Ophthalmology
645 N. Michigan Ave # 440
Chicago, IL 60611

Dear Dr. Fawzi,

Thank you for submitting your revised manuscript entitled "Müller cell glutamine metabolism links photoreceptor and endothelial injury in diabetic retinopathy". As you will see, Reviewers 1 and 3 are satisfied, while Reviewer 2 indicated some remaining concerns. We invite you to address their suggestions pertaining to Figures 4-5, in particular on the cell types that express Slc38A3, in the manner you see fit. Concerning sequencing data shown in Figure 6, please indicate in each figure legend whether data pertains to scRNA seq or bulk RNA seq to minimize the potential for reader confusion. We would be happy to publish your paper in Life Science Alliance pending these changes as well as final revisions necessary to meet our formatting guidelines.

- Please add a Summary Blurb/Alternate Abstract in our system.
- Please add a Category for your manuscript in our system.
- Please add the X and Bluesky handles of your host institute/organization, as well as your own and/or one of the authors, in our system.
- Please provide us with the clean version of the manuscript file, without highlighted text. Please remove the Significance statement from the manuscript file. You can upload this version with the file designation "Related Manuscript file".
- The contributions selected for James B. Hurley do not qualify them for authorship. Please either update the contributions in our system and in the Author Contributions section of the manuscript, or let us know if the author needs to be removed (and potentially added to the acknowledgment section).
- Please add your main, supplementary figure, and table legends to the main manuscript text after the references section.
- Please remove legends from your supplementary figures.
- Please add callouts for Figure S1A-C to your main manuscript text.
- Please add molecular weight markers to the blots in Figure 7F.
- Please ensure all images in Figure S1 contain scale bars and please indicate the scale bar size in the legend for this figure.
- Please remove Figure 9 and instead upload this image as a Graphical Abstract, with any callouts to this figure updated accordingly.

A. FINAL FILES:

B. MANUSCRIPT ORGANIZATION AND FORMATTING:

Thank you for your attention to these final processing requirements. Please revise and format the manuscript and upload materials as soon as you are able.

Sincerely,

Reviewer #1 (Comments to the Authors (Required)):

The authors have addressed my concerns.

Reviewer #2 (Comments to the Authors (Required)):

The authors have improved the manuscript extensively. However, there are still concerns remaining. The current manuscript lacks deep analysis of single-cell datasets for Müller glial and EC clusters. If the analysis is limited by the experimental process and could not provide further information, the authors may consider removing the single-cell part and focusing on the bulk RNA-seq data instead. Time course of EC and retinal cell gene change from bulk RNA-seq have provided valuable information of molecular change in early DR.

Fig. 3 is well presented and interpreted. It would be helpful to also examine metabolic genes involved in glutamine-related pathways (e.g., Slc38a3, Gls, Glud1), and mitochondrial respiration in rods and cones.

For Figs. 4-5, Slc38a3 appears to colocalize with Müller glia. Please check its expression specifically in single-cell Müller clusters, as well as in endothelial cells (Fig. 9 suggests SLC38A3 expression in MG, ECs, and photoreceptors). It would be informative to compare gene changes within Müller glial clusters. For example, Glud1 was only increased in MC_2, mitochondrial genes decreased in MC_3, and other changes were observed in MC_4, suggesting distinct metabolic states among Müller glial clusters. Which cluster do the in vitro cultured Müller glia best reflect?

It is confusing to present only a volcano plot without further analysis. Currently, the evidence supporting Müller glial glutamine metabolism impacting photoreceptors and EC health relies solely on in vitro cell culture and ex vivo retinal explant data.

Regarding Fig. 6, please clarify whether the bulk RNA-seq EC analysis can be corroborated in scRNA-seq EC clusters. Showing single-cell plots while providing bulk RNA-seq analysis is misleading. Based on the bulk RNA-seq data, in vivo ECs showed time-dependent changes. How does the in vitro culture model mimic this dynamic? Please include limitations of the in vitro work in the discussion.

Reviewer #3 (Comments to the Authors (Required)):

Authors have addressed all my concerns

We thank the Editor and the reviewers for their comments. We addressed the reviewer 2 comments as suggested by the Editor.

Reviewer 2 comments

The authors have improved the manuscript extensively. However, there are still concerns remaining. The current manuscript lacks deep analysis of single-cell datasets for Müller glial and EC clusters. If the analysis is limited by the experimental process and could not provide further information, the authors may consider removing the single-cell part and focusing on the bulk RNA-seq data instead. Time course of EC and retinal cell gene change from bulk RNA-seq have provided valuable information of molecular change in early DR.

Our single-cell RNA-seq analysis provided valuable insights, particularly for Müller glial cells, where we observed specific cluster alterations that would not have been detectable with bulk RNA-seq alone. Regarding endothelial cells, we acknowledge the limitations of our dataset, which we have clearly discussed in the manuscript. Despite these limitations, the single-cell data were still useful for comparison with the bulk RNA-seq results, helping to contextualize gene expression changes observed in early diabetic retinopathy.

Fig. 3 is well presented and interpreted. It would be helpful to also examine metabolic genes involved in glutamine-related pathways (e.g., *Slc38a3*, *Gls*, *Glud1*), and mitochondrial respiration in rods and cones.

We thank the reviewer for the suggestion. *Slc38a3* has already been analyzed in rods and cones, as shown in Figure 4A. Based on these data, we performed Western blotting on whole retinal lysates, and then immunofluorescence during the first round of revision to further investigate the cells expressing SLC38A3. The IF shows a significant decrease in the inner segment of photoreceptors. We also examined *Gls* and *Glud1* expression, which did not show significant changes neither in the cones nor in the rods.

For Figs. 4-5, *Slc38a3* appears to colocalize with Müller glia. Please check its expression specifically in single-cell Müller clusters, as well as in endothelial cells (Fig. 9 suggests SLC38A3 expression in MG, ECs, and photoreceptors).

We thank the reviewer for the comment. *Slc38a3* is indeed expressed in endothelial cells; however, as shown in the volcano plot (Figure 6C), there were no significant changes between sham and STZ conditions, indicating that the major alteration occurs in other cells (e.g. Photoreceptors). *Slc38a3* is also known to be expressed in the inner nuclear layer (INL) (Umapathy et al 2009) (also visible in our IF). Our single-cell RNA-seq did not detect significant changes in the Müller glia compartment between sham and STZ.

It would be informative to compare gene changes within Müller glial clusters. For example, Glud1 was only increased in MC_2, mitochondrial genes decreased in MC_3, and other changes were observed in MC_4, suggesting distinct metabolic states among Müller glial clusters. Which cluster do the in vitro cultured Müller glia best reflect?

We thank the reviewer for the insightful comment. We compared the different Müller glial clusters in our scRNA-seq data, and, as noted, distinct metabolic states are observed across the clusters based on different gene expression. Regarding the in vitro experiments, we cannot directly link MIO-M1 cells to a specific Müller glial cluster, as cells cultured in vitro behave differently from those in vivo. As stated in the manuscript, we carefully interpreted the in vitro data and relied primarily on ex vivo confirmation for our conclusions.

It is confusing to present only a volcano plot without further analysis. Currently, the evidence supporting Müller glial glutamine metabolism impacting photoreceptors and EC health relies solely on in vitro cell culture and ex vivo retinal explant data.

The evidence supporting Müller glial glutamine metabolism is first indicated by our scRNA-seq, which shows decreases in key genes involved in glutaminolysis, and is further corroborated by in vitro and ex vivo experiments. In addition, the alterations observed in photoreceptors and endothelial cells (scRNA-seq) are supported by immunofluorescence, Western blotting, and cell culture experiments, linking these changes to the mechanism we describe. We agree that further investigation will help clarify this mechanism in greater detail.

Regarding Fig. 6, please clarify whether the bulk RNA-seq EC analysis can be corroborated in scRNA-seq EC clusters. Showing single-cell plots while providing bulk RNA-seq analysis is misleading.

We appreciate the comment regarding Fig. 6. In the manuscript and figure legends, we confirmed to have specified whether the data originate from bulk RNA-seq or scRNA-seq, as our approach intentionally combines both assays to uncover molecular changes over time. We have ensured that each figure legend clearly indicates the type of analysis used. Regarding the specific point raised, the bulk RNA-seq endothelial cell (EC) analysis can be corroborated in scRNA-seq EC clusters. Notably, we observed mitochondrial and metabolic alterations across both datasets. Furthermore, these metabolic changes were validated in our in vitro experiments.

Based on the bulk RNA-seq data, in vivo ECs showed time-dependent changes. How does the in vitro culture model mimic this dynamic? Please include limitations of the in vitro work in the discussion.

The in vitro model was specifically designed to examine alterations observed in vivo at 3 months, focusing on glutamine deprivation under high-glucose conditions, which is relevant to mimic diabetes. While in vitro systems cannot fully recapitulate the complexity of the in vivo environment, our aim was to validate this specific aspect of

endothelial cell behavior (we have now added this limitation in the discussion section). In contrast, the bulk RNA-seq analysis spans six months of diabetes, revealing progressive changes including inflammation, vascular leakage, and metabolic alterations, culminating in endothelial cell death at six months, as expected in this mouse model. The in vitro experiments complement this by confirming one mechanistic aspect of endothelial dysfunction observed in vivo.

November 11, 2025

RE: Life Science Alliance Manuscript #LSA-2025-03434-TRR

Dr. Amani A. Fawzi
Northwestern University
Ophthalmology
645 N. Michigan Ave # 440
Chicago, IL 60611

Dear Dr. Fawzi,

Thank you for submitting your Research Article entitled "Müller cell glutamine metabolism links photoreceptor and endothelial injury in diabetic retinopathy". It is a pleasure to let you know that your manuscript is now accepted for publication in Life Science Alliance. Congratulations on this interesting work.

Your manuscript will now progress through copyediting and proofing. It is journal policy that authors provide original data upon request. During the proofing process please ensure that you modify the Data Availability section to include accession numbers for the sequencing data generated in the work. Please also add a statement to this section concerning the availability of metabolite profiling by mass spec.

DISTRIBUTION OF MATERIALS:

Again, congratulations on a very nice paper. I hope you found the review process to be constructive and are pleased with how the manuscript was handled editorially. We look forward to future exciting submissions from your lab.

Sincerely,
